# On the Stability and Scalability of Node Perturbation Learning

**Naoki Hiratani**
Harvard; Gatsby Unit, UCL
Cambridge, MA, USA
n.hiratani@gmail.com

**Yash Mehta**
Janelia; Gatsby Unit, UCL
Ashburn, VA, USA
mehtay@janelia.hhmi.org

**Timothy P. Lillicrap**
Deepmind
London, UK
countzero@google.com

**Peter E. Latham**
Gatsby Unit, UCL
London, UK
pel@gatsby.ucl.ac.uk

## Abstract

To survive, animals must adapt synaptic weights based on external stimuli and rewards. And they must do so using local, biologically plausible, learning rules – a highly nontrivial constraint. One possible approach is to perturb neural activity (or use intrinsic, ongoing noise to perturb it), determine whether performance increases or decreases, and use that information to adjust the weights. This algorithm – known as node perturbation – has been shown to work on simple problems, but little is known about either its stability or its scalability with respect to network size. We investigate these issues both analytically, in deep linear networks, and numerically, in deep nonlinear ones. We show analytically that in deep linear networks with one hidden layer, both learning time and performance depend very weakly on hidden layer size. However, unlike stochastic gradient descent, when there is model mismatch between the student and teacher networks, node perturbation is always unstable. The instability is triggered by weight diffusion, which eventually leads to very large weights. This instability can be suppressed by weight normalization, at the cost of bias in the learning rule. We confirm numerically that a similar instability, and to a lesser extent scalability, exist in deep nonlinear networks trained on both a motor control task and image classification tasks. Our study highlights the limitations and potential of node perturbation as a biologically plausible learning rule in the brain.

## 1   Introduction

The immense success of deep learning in recent years has revived interest in the backpropagation algorithm (known simply as "backprop") as a learning mechanism in the brain [1, 2]. Although in its pure form this algorithm is incompatible with biological constraints (most notably, the fact that synapses do not know the weights of other synapses in the brain), many biologically plausible approximations have been proposed [3, 4, 5, 6, 7, 8, 9]. However, it remains unclear how robust these approximate learning rules are, or how they perform on large neural networks [10], partially because we do not have a clear analytical understanding of them.

In the node perturbation (NP) algorithm, synaptic weights are updated according to how the error changes when a small perturbation is added to each node of a neural network [11, 4, 12]. If a perturbation improves performance, the weights are modified so that the perturbation is consolidated, and vice versa (see §3). This algorithm has two advantages over more standard biologically plausible

36th Conference on Neural Information Processing Systems (NeurIPS 2022).

learning rules. First, synaptic updates rely only on a global error signal and pre- and postsynaptic activity. Such a global error signal, mediated by neuromodulators, is commonly observed across the brain [13], and is known to modulate synaptic plasticity [14]. This is in contrast to many biologically plausible credit assignment algorithms, which require vector-valued supervising signals (what the true label was), not just scalar error signals (how accurate the network output was). Second, neural activity in the brain is inherently stochastic [15, 16], which NP can make use of – making noise a feature rather than a bug.

Because of its biological plausibility and algorithmic simplicity, NP has been investigated as a potential learning mechanism of several brain regions, including the vocal production circuits of songbirds [17, 18], motor cortex [19], and the cerebellum [20]. In particular, experimental work on birdsong learning suggest that perturbation of neural activity in the song production circuit via a projection from a region called LMAN is crucial for song acquisition [21, 22], supporting the presence of perturbation-based learning.

Here we analyze the learning dynamics of node perturbation. We develop a mean-field theory of NP learning in one hidden layer linear networks, and we show analytically, and confirm empirically, that the minimum training time of NP depends linearly on the number of output nodes, as suggested previously [12]. Perhaps surprisingly, we show that the training time is independent of the hidden layer width in the over-parameterized regime. Finally, we show that NP is always unstable when the supervised signal contains noise. This instability can be mitigated by regularizing the weight norms, but that induces bias in the update.

To verify that these results are not specific to deep linear networks, we investigate the scalability and stability of NP in nonlinear deep neural networks applied to the SARCOS [23] and MNIST tasks [24]. In both tasks, the minimum learning time required to reach high performance scales sub-linearly with the hidden layer size. These observations support the relative robustness of NP against hidden layer width expansion, a prediction from linear networks. However, NP dynamics is unstable at high learning rate, with the instability preceded by an expansion of the weight norm, as predicted. Weight normalization prevents this instability, but impairs performance, especially for SARCOS. For MNIST, stabilization by weight normalization improves the convergence time of NP against the depth expansion. We also confirmed the stabilization of learning dynamics by weight regularization in convolutional neural networks solving the CIFAR-10 task [25], but it also impairs performance.

The minimum learning time of NP is about one hundred times longer than that of stochastic gradient descent (SGD) in both tasks. However, a portion of this discrepancy is explained by the fact that NP is a reinforcement learning algorithm while SGD uses supervising signals. Our study thus suggests that, depending on the task and the architecture, NP may play a role as a credit assignment mechanism in the brain. However, our results also indicate that NP alone is not enough to account for the learning ability of the brain. In particular, NP is too slow to be practical in supervised image recognition tasks.

## 2   Related work

NP has been previously proposed as a learning mechanism of birdsong [17, 18], motor cortex [19], and cerebellum [20]. Nonetheless, the practicality of NP as a credit assignment mechanism in the brain remains unclear, as all of these studies considered simplified tasks, and scarcely provided analytical insight. Werfel and colleagues did provide analytical insight into the scalability of NP in a linear regression setting: they showed that the convergence of NP becomes progressively slower than SGD as the output layer size increases [12]. However, they could not tell whether the slower convergence was due to an increase in the number of perturbed neurons or the number of output neurons, as they were the same. In this work, we distinguish these two possibilities by introducing a hidden layer (see §4 for the details). Fiete and colleagues numerically demonstrated that when the loss function is defined on a low dimensional projection of output neurons, the training time doesn't depend on the output layer size [17]. However, it remains unclear how general this result is, as their analysis is limited to a linear regression setting. Lansdell and colleagues used NP to train feedback weights of a deep neural network that back-propagates the error [26]. The performance of their algorithm was comparable to SGD on CIFAR-10. However, it required supervising signals, unlike the vanilla NP, which only needs reinforcement signals. In addition, it remains unclear how much performance gain in their algorithm was brought by the weight alignment rather than from NP.

If perturbations are added to synaptic weights instead of neural activity, the algorithm is called weight perturbation [3]. This algorithm is noiser than node perturbation in the vanilla setting [12], but a recent result suggests that the weight perturbation algorithm is competitive with, or better than, NP when the feedback signal is sparse [27]. It should also be noted that weight perturbation is a variation of zeroth order optimization, which is known to scale badly to high-dimensional optimization problems [28, 29]. However, recent works found that when there is a low-dimensional latent structure in the parameter space, this algorithm works efficiently [30, 31].

Scalability of noisy weight update rules has also been investigated in the bandit literature [32]. In bandit problems, the gradient needs to be estimated from insufficient feedback, thus its estimation inevitably becomes either biased or noisy. This limitation raises the question of whether one should choose a biased update or a noisy one, for which the following property is known [33]. Consider an optimization of a strongly convex loss function $L(w)$ with a biased noisy gradient. Suppose there is a parameter $\delta$ such that, the update rule is noisy but unbiased in the limit $\delta \to 0$, and biased but noise free in the limit $\delta \to \infty$. Then, under some mild conditions, the value of $\delta$ that achieves the minimax regret under a fixed number of iterations $n$, denoted $\delta^*$, is given by $\delta^* \propto \sqrt{\frac{cd}{n}}$ where $d$ is the number of parameters. Consequently, when the number of iterations $n$ is small while the number of parameters $d$ is large, as in the brain, the system should choose a biased update, not a noisy one. However, the results from convex optimization are often not directly applicable to deep neural networks, because the effective number of degrees of freedom is typically much smaller than the number of parameters in these networks [34], and the loss function is non-convex. Thus, it remains unclear if the brain should always choose a biased learning rule over a noisy one, such as NP.

## 3   Node perturbation is unbiased, but noisy

We formulate node perturbation in the context of deep learning. Let us consider a vanilla deep feedforward network

$$\boldsymbol{x}_k = f_k(\boldsymbol{W}_k \boldsymbol{x}_{k-1}),\ k = 1, 2, ..., K \tag{1}$$

where $\boldsymbol{x}_0$ and $\boldsymbol{x}_K$ are the input and the output respectively, and $f_k(\cdot)$ is an element-wise activation function. Throughout the paper, we use bold italic lower case letters to denote column vectors and bold italic upper case letters to denote matrices. Adding a small Gaussian perturbation $\sigma\boldsymbol{\xi}_k$ to each layer except the input layer gives us a perturbed network,

$$\widetilde{\boldsymbol{x}}_k = f_k(\boldsymbol{W}_k \widetilde{\boldsymbol{x}}_{k-1} + \sigma\boldsymbol{\xi}_k),\ k = 1, 2, ..., K \tag{2}$$

where $\langle \boldsymbol{\xi}_k \boldsymbol{\xi}_l^T \rangle = \delta_{kl} \boldsymbol{I}_k$, with $\boldsymbol{I}_k$ the identity matrix in the appropriate dimension, and $\sigma \ll 1$. Under a loss function $\ell(\boldsymbol{x}_K, \boldsymbol{x}_0)$, the node perturbation update is

$$\delta\boldsymbol{W}_k^{np} = -\frac{\eta}{\sigma} \left( \ell(\widetilde{\boldsymbol{x}}_K, \boldsymbol{x}_0) - \ell(\boldsymbol{x}_K, \boldsymbol{x}_0) \right) \boldsymbol{\xi}_k \boldsymbol{x}_{k-1}^T. \tag{3}$$

In words, if the perturbation decreases the error (i.e., $\tilde{\ell} - \ell < 0$), the weights are shifted towards the direction of the perturbation ($\boldsymbol{\xi}_k$), and vice versa. Importantly, in this update rule, the network only needs to know how much the loss changes when a perturbation is added to the network. This is in contrast to SGD and most of its biologically plausible variants, which require a supervised signal telling what the correct answer was. It is straightforward to show that, at the small perturbation limit (ie. $\sigma \to 0$; see Appendix A.1),

$$\delta\boldsymbol{W}_k^{np} = -\eta \sum_{l=1}^{K} \boldsymbol{\xi}_k \boldsymbol{\xi}_l^T \boldsymbol{g}_l \boldsymbol{x}_{k-1}^T, \quad \text{where} \quad \boldsymbol{g}_k \equiv \frac{\partial \ell}{\partial \boldsymbol{h}_k}, \quad \text{and} \quad \boldsymbol{h}_k \equiv \boldsymbol{W}_k \boldsymbol{x}_{k-1}. \tag{4}$$

Taking the expectation over $\boldsymbol{\xi}$, we recover the SGD update rule,

$$\langle \delta\boldsymbol{W}_k^{np} \rangle_{\boldsymbol{\xi}} = -\eta \boldsymbol{g}_k \boldsymbol{x}_{k-1}^T = \delta\boldsymbol{W}_k^{sgd}. \tag{5}$$

Consequently, NP is unbiased relative to SGD. However, because the weight update is driven by random perturbation to the nodes, NP is much more noisy than SGD. For instance, the cosine similarity between SGD and NP update at $k$-th layer is given by

$$\left\langle \cos\left( \angle\left[ \delta\boldsymbol{W}_k^{sgd}, \delta\boldsymbol{W}_k^{np} \right] \right) \right\rangle_{\boldsymbol{x},\boldsymbol{\xi}} \lesssim \frac{1}{\sqrt{L_k}}, \tag{6}$$

where $L_k$ is the number of neurons in the $k$-th layer (see Appendix A.2). Thus, for wide networks, the NP and SGD updates becomes nearly perpendicular (see Figs. 4A and S4A).

Another way to characterize how noisy NP is to look at the noise covariance. Because both SGD and NP are unbiased against the true gradient, both update rules can be written as

$$\delta \boldsymbol{W}_k^Q = -\eta \left( \left\langle \boldsymbol{g}_k \boldsymbol{x}_{k-1}^T \right\rangle_{\boldsymbol{x}} + \boldsymbol{Z}_k^Q \right), \tag{7}$$

where $Q = np$ or $sgd$, and $\boldsymbol{Z}_k^Q$ is a (not necessarily Gaussian) random matrix. We show in Appendix A.3 that the covariance of $\boldsymbol{Z}_k^Q$ under NP, denoted $C_{kl}^{np}$, is written as,

$$C_{kl}^{np} \approx 2C_{kl}^{sgd} + \delta_{kl} \left\langle \sum_{m=1}^{K} \|\boldsymbol{g}_m\|^2 \boldsymbol{I}_k \otimes \boldsymbol{x}_{k-1} \boldsymbol{x}_{k-1}^T \right\rangle_{\boldsymbol{x}}, \tag{8}$$

where, $\boldsymbol{I}_k$ is a size-$L_k$ identity matrix, $C_{kl}^{sgd}$ is the noise-covariance under SGD, and "$\otimes$" denotes a tensor product between two matrices (i.e., $[A \otimes B]_{ijkl} = A_{ij}B_{kl}$). Because the second term is zero unless two weights share the same postsynaptic neuron, the noise correlation of NP updates across neurons in different layers, or across different neurons in the same layer, is only a factor of two larger than that of SGD. However, the NP update at each synapse contains additional, mostly independent, noise from the second term that scales with the total gradient norm $\sum_{m=1}^{K} \|\boldsymbol{g}_m\|^2$. This term is typically much larger than the SGD variance, which only depends on the gradient amplitude at each weight (see Eq. 30).

## 4   NP in one hidden layer linear networks: scalable but unstable

We have so far observed that NP is unbiased but noisy, meaning that a one-step update by NP is just a noisy version of SGD. However, it is not clear what that implies for the global stability and the scalability of the algorithm. Because it is very difficult to address these issues in general, here we first focus on deep linear networks in a student-teacher setting [35, 36].

In the absence of hidden layers (i.e., in a linear regression setting), it is known that the minimum number of training steps required to reach a target error level $\epsilon_{tg}$ from an initial error $\epsilon_o$ is

$$T_{sgd}[\epsilon_o \rightarrow \epsilon_{tg}] = L_x \log(\epsilon_o/\epsilon_{tg}), \quad T_{np}[\epsilon_o \rightarrow \epsilon_{tg}] = L_x L_y \log(\epsilon_o/\epsilon_{tg}), \tag{9}$$

where $L_x$ and $L_y$ are input and output layer sizes, respectively (see [12] and Appendix B.6). Thus, NP is linearly slower than SGD as the number of output units increases. Note that at a small learning rate $\eta$, the learning trajectories of NP and SGD behave similarly (left edge of Fig. S1, in the Appendix). However, the optimal learning rate – the learning rate that minimizes the training time – is smaller under NP than SGD, so the minimum training time becomes longer (Fig. S1; here, the optimal learning rates are the points at which the curves touch the horizontal dotted lines). Nevertheless, in the linear regression setting, NP learning dynamics is simple, and monotonic: the error either decreases or increases monotonically throughout learning. As we will see, however, this is not the case in deep neural network; they display non-trivial, and non-monotonic, dynamics.

Equation (9) tells us that NP is slower than SGD by a factor of $L_y$, the number of output units. However, it doesn't tell us in general whether training time scales with the number of output units or the number of perturbed units, because the two are the same in linear regression. To disambiguate between these two possibilities, we analyze a linear network with one hidden layer,

$$\boldsymbol{y} = \boldsymbol{W}_2 \boldsymbol{W}_1 \boldsymbol{x}, \tag{10}$$

where the matrices $\boldsymbol{W}_2$ and $\boldsymbol{W}_1$ are $L_y \times L_h$ and $L_h \times L_x$, respectively. We consider a student-teacher setting, in which the target output $\boldsymbol{y}^*$ is generated from a teacher network,

$$\boldsymbol{y}^* = \boldsymbol{A}\boldsymbol{x} + \sigma_t \boldsymbol{\zeta}. \tag{11}$$

where $\sigma_t \boldsymbol{\zeta}$ is zero mean Gaussian noise that creates a mismatch between the student and the teacher networks. We implement NP in this network by applying small perturbations to both the hidden and output layers simultaneously (see Eq. 3 and Appendix B.1). Although the exact learning dynamics is not tractable for this network, in the limit of large hidden layer size, and under the assumption that weight updates are dominated by noise, we can use a mean-field approximation to

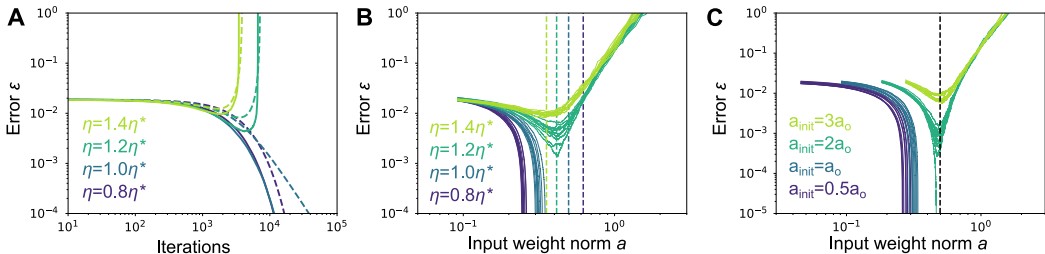

Figure 1: **Learning dynamics of NP in one hidden layer linear networks. A)** Learning dynamics of node perturbation under various learning rates. Solid lines are empirical results and dotted lines are theory (Eq. 12). $\eta^*$ is the analytically-estimated critical learning rate. **B)** The same as **A**, but plotted in the phase space spanned by the input weight norm and the error. Lines represent trajectories with different random seeds. Dotted vertical lines are the threshold $2/c_o$. **C)** Learning dynamics under different initial weight norms. In all three panels, we set $L_x = 100$, $L_h = 10000$, $L_y = 10$, and $\sigma_t^2 = 0$. See Appendix C.2 for further details. Simulation codes for the figures are made available at https://github.com/nhiratani/node_perturbation.

gain considerable insight into the learning dynamics as described below. Our derivation proceeds in two steps. First, we introduce four order parameters capturing the weight norms $\|W_1\|_F^2$ and $\|W_2\|_F^2$, error $\|W_2 W_1 - A\|_F^2$, and their inner product $\text{tr}[(W_2 W_1 - A)^T W_2 W_1]$, and approximately write down the learning dynamics in terms of these order parameters (B.2). Secondly, we reparameterize the learning dynamics, which leads us to two-variable description of the dynamics (B.3). The derivation of the mean-field approximation is detailed in Appendix B.

**Scalability of NP at the zero teacher noise limit.** We first consider zero teacher noise ($\sigma_t = 0$ in Eq. 11). In a network satisfying $L_h \gg L_x \gg 1$ and $L_h \gg L_y \gg 1$, and using the Xavier-Glorot initialization (see [37] and Eq. 69), for the first $\mathcal{O}(L_h^{1/3})$ iterations, the learning dynamics of NP is described by the following coupled dynamics (see Appendix B.3),

$$\dot{a} = c_o \epsilon, \quad \dot{\epsilon} = -(2 - c_o a) a \epsilon, \tag{12}$$

where $\epsilon \equiv \frac{1}{L_x L_y} \|W_2 W_1 - A\|_F^2$ is the loss, $a \equiv \frac{1}{L_x L_h^{1/3}} \|W_1\|_F^2$ is the normalized input weight norm, and $c_o$ is a positive coefficient proportional to the learning rate (see Eq. 76). This simple dynamical system captures the learning dynamics of NP well (Fig. 1A; solid lines are simulations, and dotted lines are Eq. 12). Because $\epsilon \geq 0$ by definition, the norm $a$ increases monotonically with time. By contrast, the loss $\epsilon$ either decreases or increases depending on whether $a$ is smaller than $2/c_o$ or not. When the learning rate $\eta$ is fixed to a small value, $c_o$ is also small. Therefore, the weight norm $a$ increases slowly (purple and blue lines in Fig. 1B), and the weight norm threshold $2/c_o$ is relatively large (purple and blue vertical dotted lines). Thus, the error converges to zero before the weight norm crosses the threshold. In contrast, when the learning rate is larger than the critical value $\eta^*$ (defined by Eq. 83), the weight norm surpasses the threshold $2/c_o$ before the error converges to zero; at that point the error starts to go up again (green lines in Fig. 1B). Similarly, if the initial weight norm is set to a large value, the error rises before converging to zero (green vs. blue lines in Fig. 1C). This non-monotonic learning dynamics is in contrast to the linear regression setting in which the error either decreases or increases monotonically throughout learning.

When we optimize the learning rate, we find that the minimum number of training steps required to reach a target error level, $\epsilon_{tg}$, from an initial error, $\epsilon_o$, under SGD and NP are given by

$$T_{sgd}[\epsilon_o \to \epsilon_{tg}] = L_x \log(\epsilon_o/\epsilon_{tg}), \quad T_{np}[\epsilon_o \to \epsilon_{tg}] = L_x L_y \frac{1 + \beta_o}{c_o} \int_{a_o}^{a_{tg}} \frac{da}{c_o(\epsilon_{tot} + a^3/3) - a^2}, \tag{13}$$

where $\beta_o, c_o, \epsilon_{tot}$ are $\mathcal{O}(1)$ coefficients, and $a_o$ and $a_{tg}$ are the weight norms under the initial error $\epsilon_o$ and the target error $\epsilon_{tg}$, respectively (see Appendix B.3). Notably, the training time $T_{np}$ doesn't depend on the hidden layer size $L_h$ except for small-size effects on $a_o$ and $\beta_o$ (see Eq. 71), indicating

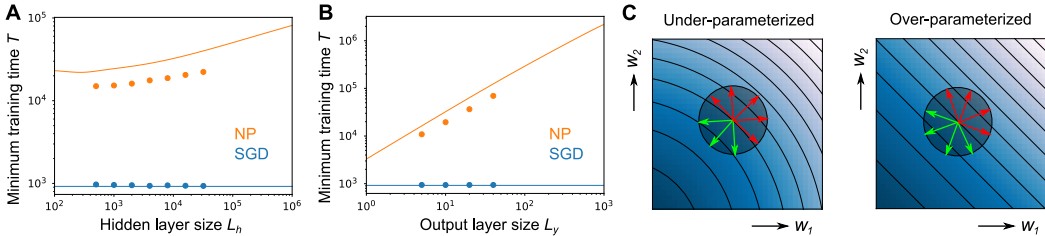

Figure 2: **Scalability of NP in one hidden layer linear networks. A, B**) Minimum training time of SGD (blue) and NP (orange) under a range of hidden (**A**) and output (**B**) layer sizes, in a linear network with one hidden layer. Lines are theory (Eq. 13); points are empirical estimations. In panel **A**, $L_y = 10$; in panel **B**, $L_h = 10000$. We used $L_x = 100$ and $\sigma_t = 0.0$ in both. **C**) Schematic figures of a weight update under NP. Blue shade represents the loss landscape (blue: low loss, white: high loss), and green and red arrows represent good and bad directions, respectively.

that NP doesn't slow down linearly with the number of perturbed nodes. Indeed, simulations show that the minimum training time is almost flat when plotted against the hidden layer size (Figs. 2A and S2A). Moreover, the optimal learning rate in the simulations scales as $\eta \propto L_h^{-1/3}$, as predicted (Fig. S2B). In contrast, the training time increases linearly with the output layer size $L_y$ (Fig. 2B).

It may appear puzzling that training time saturates in the large hidden layer size limit (Fig. 2A and S2A), considering that weight updates under NP are almost random in that limit (Eqs. 6 and 8). However, as illustrated in Fig. 2C, while the noise in the weight updates (gray circles) increases the loss for some directions (red arrows), it decreases it for others (green arrows). When the loss function is strongly convex, there are always more bad directions (red arrows) than good directions (green arrows), as illustrated in the left panel of Fig. 2C. Therefore, and also because the loss associated with red arrows is larger than the gain with green arrows in a convex loss, noise is detrimental to learning on average. Nevertheless, in overparameterized networks, there are a large number of directions that don't affect the loss (because the Hessian of an over-parameterized neural network is typically low rank [34]); such directions are shown in the right panel of Fig. 2C. Since NP noise is mostly homogeneous, the more overparameterized the network is, the lower the volume of weights space in which the loss increases. Thus, in an over-parameterized network, NP noise is not as detrimental as anticipated from convex optimization theory.

**Instability of NP in the presence of teacher noise.** So far we have focused on the zero-noise limit ($\sigma_t = 0$). We now investigate the effect of non-zero noise ($\sigma_t \neq 0$) in the teacher network, which introduces a mismatch between the student and the teacher networks. Our goal is to understand how this mismatch affects the stability of the learning dynamics. In this setting, the error $\epsilon$ decreases only when (see Eq. 84a)

$$\epsilon > \frac{c_o a \sigma_t^2}{L_x (2 - c_o a)}. \tag{14}$$

Thus, even at a very small learning rate (proportional to $c_0$), eventually Eq. 14 will not be satisfied, and the error will begin to increase. This process is illustrated in Fig. 3A: the loss decreases until it crosses the dotted lines, which are the nullclines, given by equality in Eq. 14: $\epsilon = c_o a \sigma_t^2 / (L_x (2 - c_o a))$. Because weight norm $a$ is a monotonically increasing function of time, the right-hand side of Eq. 14 is also monotonically increasing. Thus, this instability occurs faster under a larger learning rate (green vs blue lines in Fig. 3A).

The black line in Fig. 3B shows the number of iterations it takes until the error starts to increase. At a learning rate $\eta > \eta^*$, the number of iterations scales roughly as $\eta^{-3}$ (right side of vertical line), but the scaling becomes more moderate ($\propto \eta^{-1.5}$) at a lower learning rate (left side). Moreover, the error eventually surpasses the initial error level, though that takes a long time when the learning rate is below the critical value (gray line). Under SGD, the teacher noise affects the learning dynamics in a qualitatively different manner. Up to a certain learning rate, SGD dynamics is always unstable from the initial update (filled circles in Fig. 3C), whereas below the critical learning rate, the error keeps decreasing almost monotonically at least for $10^6$ iterations (open circles).

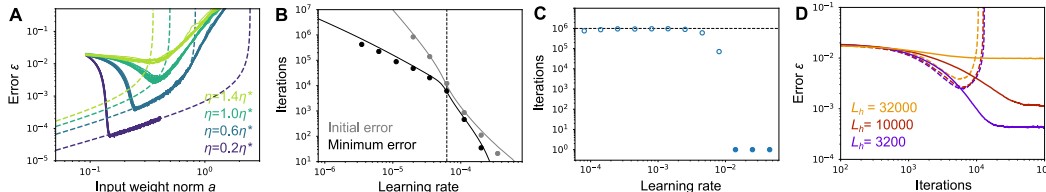

Figure 3: **Instability of NP in one hidden layer linear networks with teacher noise. A)** Learning dynamics of node perturbation in the presence of teacher noise. Lines are simulations, dotted lines are Eq. 14. **B)** The number of iterations until the error reaches a minimum and starts to increase (black) and until the error goes back to its initial value (gray). Points are simulations and lines are theory calculated from Eq. 84a. The vertical dashed lines is $\eta^*$. **C)** The number of iterations until the error reaches a minimum under SGD. The dashed horizontal line represents the iteration where simulation was terminated, meaning that on average the error decreased monotonically for at least $10^6$ iterations. **D)** Learning dynamics under three different hidden layer sizes with (solid lines) and without (dotted lines) weight normalization. The learning rates were set to the critical rate $\eta^*$ under $\sigma_t = 0$ (Eq. 83). The dotted lines are almost identical to each other due to the scale invariance. In all panels we used $L_x = 100$, $L_y = 10$, and $\sigma_t^2 = 0.1$. In panels **A-C**, we used $L_h = 10000$.

Our analysis so far indicates that the instability of NP is triggered by the expansion of the weight norm. Thus, we expect the system to regain stability if we regularize the weight. Indeed, if we keep the norm of the incoming weights of each neuron constant, the error converges – unlike for the un-regularized network (Fig. 3D; solid vs. dotted lines). However, because the weight normalization induces bias in the weight update, the error no longer goes to zero; instead, it saturates at a finite value. The effect of this bias is particularly detrimental when the hidden layer size is large, because strong weight normalization is necessary for a larger network (yellow vs purple lines in Fig. 3D).

## 5 NP in nonlinear networks solving motor and visual tasks

Our analysis in deep linear networks revealed that NP scales well against over-parameterization, but the learning dynamics is susceptible to an instability because noise expands the weight norm monotonically. Weight normalization suppresses this instability, but the normalization also introduces a bias that impairs scalability. To see if these results generalize to more complex tasks that better mimic typical credit assignment problems the brain faces, we analyze nonlinear networks solving the SARCOS [23], MNIST [24], and CIFAR-10 tasks [25].

We first applied NP to the SARCOS dataset, a kinematics dataset collected from a seven degree-of-freedom SARCOS robotic arm [23, 38, 39]. The task is to predict the torques at the joints (seven-dimensional output) given their positions, velocities, and accelerations as inputs (21-dimensional input). We chose this task because NP is previously proposed as a learning algorithm for motor systems [17, 19, 20]. As expected, NP is indeed noisy, even with mini-batch implementation, resulting a low cosine similarity with SGD (Fig. 4A; here mini-batch size is 100). In particular, in one-hidden layer networks the cosine similarity between the NP and SGD weight updates in the hidden layer fall off as $1/\sqrt{L_h}$, while the similarity in the output layer stays roughly constant, as predicted by the theory (Fig. 4A and Eq. 6).

Moreover, learning under NP is indeed susceptible to an instability, as predicted. Under a small learning rate, the learning dynamics of NP is stable for more than $10^4$ epochs (purple and blue lines in Fig. 4B), but as the learning rate increases, the error starts to diverge (green lines). Because the weight norm increases monotonically, we expect NP dynamics to eventually become unstable even under a small learning rate, though it is difficult to confirm numerically due to the high computational cost. The learning-rate dependence of this instability is also similar to what we observed in deep linear networks (compare Fig. 4C with Fig. 3B). At a relatively high learning rate, the error, measured by the test error, surpasses the initial error level immediately after the error starts to go up (right half of Fig. 4C; black and gray lines are close to each other). On the other hand, at a lower learning rate, the error increases slowly after it reaches the minimum (left half of Fig. 4C). This instability is absent from SGD learning; in that case, learning is either stable or unstable from the beginning (Fig. S3A).

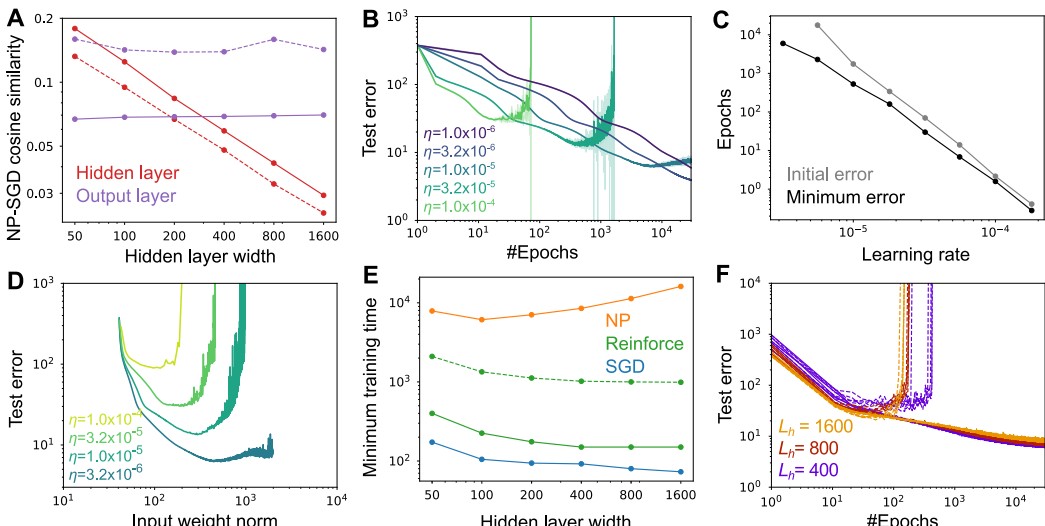

Figure 4: **Nonlinear networks solving SARCOS. A**) Cosine similarity between NP and SGD at various hidden layer widths. Solid and dotted lines are the similarities around the beginning ($\epsilon = 50.0$) and the end ($\epsilon = 5.0$) of learning, respectively. **B**) Learning dynamics of NP under various learning rates. Shades, which are barely visible at small learning rates, are the standard deviations over five random seeds. **C**) The number of epochs until the error starts to increase (black) and the error goes back to the initial error level (gray). **D**) Learning dynamics of NP in a space spanned by the input weight norm and the test error measured in MSE. **E**) Minimum training time of NP, Reinforce, and SGD in one hidden layered networks with various hidden layer widths. In dashed green line, we added perturbation to both the intermediate and output layers. **F**) NP learning with (solid) or without (dotted) weight normalization. See Appendix C.2 for the details.

Plotting the test error against the squared weight norm of the input layer $\|\boldsymbol{W}_1\|_F^2$, we see U-shaped curves as in deep linear networks (Fig. 4D vs. Fig. 1B and 3A). This result suggests that expansion of the input weight norm underlies the observed NP instability, even in nonlinear networks.

We next estimated the scalability of NP against the hidden layer width. Under NP, the minimum training time required for reaching the target (here set to the test error, $\epsilon = 5.0$, which corresponds to $R^2 = 0.986$, roughly one hundredth of the initial error) depends only weakly on the hidden layer width (Fig. 4E vs. Fig. 2A; see Fig. S3B for the choice of the optimal learning rates). However, we also observed that its learning efficiency is about one hundred times worse than SGD (orange vs. blue lines in Fig. 4E). One key difference between NP and SGD, which may partially explain this discrepancy, is that NP is a reinforcement learning algorithm that doesn't use a vector-valued supervised signal, unlike SGD. To examine its effect, we consider the following reinforcement learning algorithm [40]. By adding perturbations only to the output layer, the gradient with respect to the last layer is estimated as

$$\boldsymbol{g}_K \approx \widetilde{\boldsymbol{g}}_K = (\ell(\widetilde{\boldsymbol{x}}_K, \boldsymbol{x}_o) - \ell(\boldsymbol{x}_K, \boldsymbol{x}_o)) \, \boldsymbol{\xi}_K / \sigma. \tag{15}$$

By back-propagating the gradient, $\widetilde{\boldsymbol{g}}_K$, the network can approximately perform SGD based on the reinforcement signals $\ell(\widetilde{\boldsymbol{x}}_K)$ and $\ell(\boldsymbol{x}_K)$. The minimum training time under this reinforcement learning algorithm is about three times longer than SGD (solid green versus blue lines in Fig. 4E), though much better than NP. Moreover, assuming that the perturbation originated from intrinsic noise in the system, we added perturbations to the hidden layer as well. This additional perturbation impairs the performance of the reinforcement learning algorithm (dashed vs solid green lines). However, NP is still five to ten times slower than this noisy reinforcement learning (orange vs. dashed green lines).

Our analysis of deep linear networks indicates that weight normalization prevents the instability in NP learning dynamics. Indeed, in the presence of neuron-wise weight normalization (see Appendix C.1), learning dynamics becomes stable even when the hidden layer size is large (dotted vs solid lines in Fig. 4F), but this prevents the error from converging to the target level ($\epsilon = 5.0$).

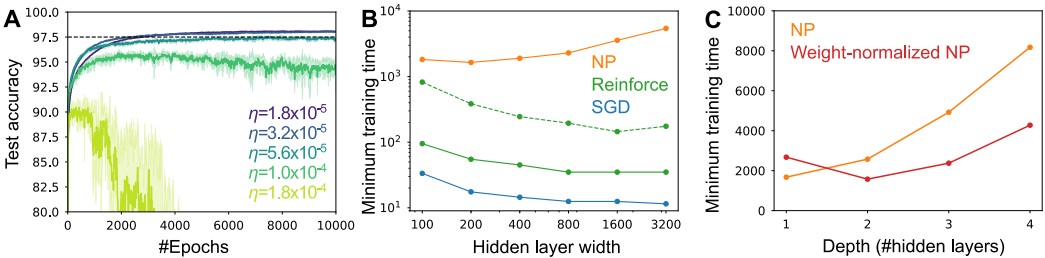

Figure 5: **Nonlinear networks solving MNIST. A**) Learning dynamics of NP under various learning rates. Shades represent the standard deviation over 5 random seeds. **B**) Minimum training time of NP, Reinforce, and SGD in one hidden layered networks at various hidden layer widths. Solid and dashed green lines are reinforcement learning without/with additional perturbation to the hidden layers, respectively. **C**) Minimum training time of NP with or without weight normalization in networks with various depths. Here, we set all the hidden layer sizes to 300 regardless of the depth.

At a low learning rate, NP and SGD show similar error curves (Fig. S3C), but it doesn't mean that the two algorithms learn the same representation. We found that the linear dimensionality of the hidden layer representation, measured by the participation ratio of PCA eigenvalues (see Appendix C.1), was lower under NP than SGD, even at a low learning rate (Fig. S3D). This result implies that it may be possible to distinguish the two learning rules, just by observing activity in the hidden layer.

NP exhibits similar learning dynamics when applied to MNIST (Figs. 5 and S4). Under a large learning rate, learning becomes unstable before reaching high performance (green lines in Fig. 5A). The minimum training time depends only sub-linearly on the hidden layer width (orange line in Fig. 5B). Compared to SGD, NP learning is one hundred to one thousand times slower, but the performance deficit against reinforcement learning is smaller (Fig. 5B). Although the analysis of the depth dependence is complicated even in the linear setting, we empirically observed that the minimum learning time of NP scales supra-linearly with respect to depth (orange line in Fig. 5C). However, this effect can be ameliorated by introducing weight normalization (red line). Lastly, we applied NP to a convolutional network learning CIFAR-10, and found that weight regularization stabilizes NP learning dynamics (Fig. S5). However, it only achieves a low accuracy ($< 50\%$) after a few thousand epochs of training due to its bias and slowness.

# 6 Discussion

We analyzed the learning dynamics of NP in deep neural networks and revealed its limitation and potential. This study provides several insights into biologically plausible learning mechanisms.

Firstly, considering biological implementation, it is often useful to measure the performance by the minimum number of iterations required to reach a target – which could, for instance, be human-level performance. This is because an algorithm is biologically implausible if it takes an inordinately long time to reach good performance. For instance, under a sufficiently small learning rate, NP eventually achieves good performance, since it is unbiased (§3). However, convergence is significantly slower than SGD (§4 and §5), which limits its utility, especially in the presence of supervising signals. Though we found the minimum training time of NP is relatively robust against over-parameterization, its scalability against complex supervised learning tasks is clearly limited due to this slowness.

In the brain, learning often needs to rely on scalar-valued reward signals rather than supervised signals. Although NP is much slower than SGD, its performance deficit is smaller, when compared to a reinforcement learning rule using the error backpropagation, though NP is still more than one order of magnitude slower (Figs. 4E and 5B). Thus, NP might be utilized in a noisy reinforcement learning problem, such as motor learning, as a building block of learning mechanism.

Our work highlights the importance of theoretical investigations in an idealized scenario, such as a deep linear network, not only for understanding how the proposed learning algorithm works, but also for obtaining insight into what kind of regularization is needed for making a learning algorithm works efficiently. We revealed that neuron-wise weight normalization stabilizes the learning dynamics. By

contrast, a constant-rate weight decay doesn't work robustly, because the change in the weight norm is not constant (black vs colored lines in Fig. S4F).

Finally, it is important to analyze the representation in the hidden layers in the learned circuit for obtaining an experimentally testable prediction [41, 42]. Although the representation in the output layer is almost invariant among any successful learning rules by definition, hidden layer representation may vary depending on the learning rule, particularly in over-parameterized neural networks. Here, we showed that the linear dimensionality of the hidden layer representation is different between NP and SGD, even when they achieve similar performance (Figs. S3CD and S4E).

## Acknowledgement

This work was supported by the Swartz Foundation (NH), Wellcome Trust (110114/Z/15/Z; PEL), and the Gatsby Charitable Foundation (PEL).

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
