# Supplementary Materials for On the Stability and Scalability of Node Perturbation Learning

Naoki Hiratani[1], Yash Mehta, Timothy P. Lillicrap, Peter E. Latham

## Supplementary Figures

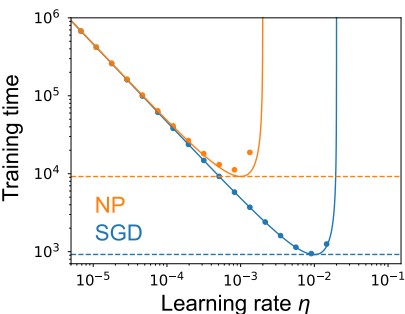

Figure S1: **Linear regression model.** The number of updates required for reducing the error to 1/10000 of the initial error in linear regression setting. Dotted horizontal lines are Eq. 9.

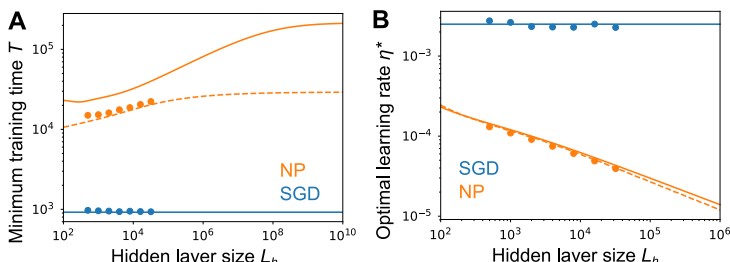

Figure S2: **Scalability of NP in deep linear networks. A**) Same as Fig. 2A, but plotted in a wider parameter range. Solid lines are the closed form solutions (Eq. 13), while the orange dotted line was obtained by numerically solving the mean-field dynamics (Eq. 68). Points are empirical estimations. Simulations results were (obtained only for small $L_h$, due to high computational cost for large hidden layer size). **B**) The optimal learning rates – the rates that achieve the minimum training time depicted in Fig. S2A and Fig. 2A. As in panel **A**, solid lines are closed-form solutions, and the orange dotted line is obtained by numerically solving the mean-field dynamics.

[1]n.hiratani@gmail.com

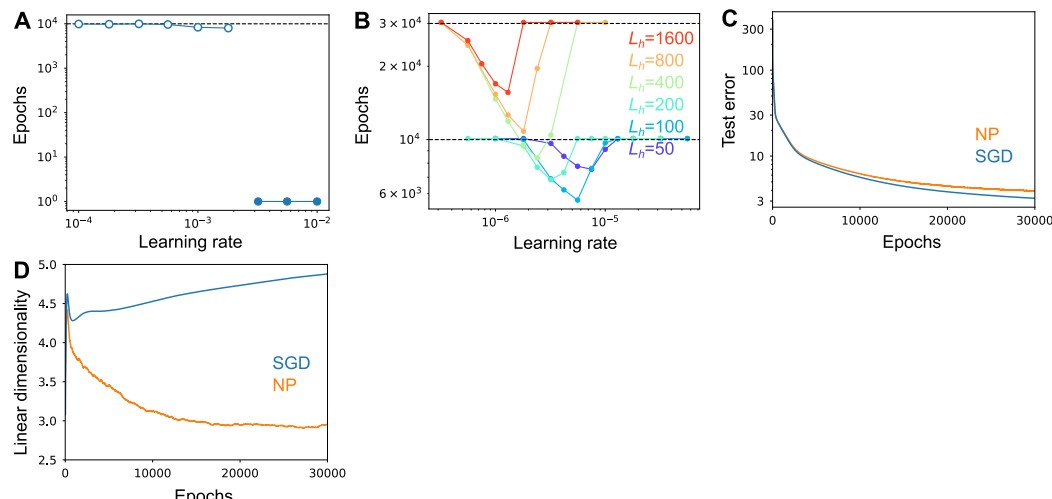

Figure S3: **Nonlinear networks solving SARCOS. A)** The number of epochs under SGD, until either the error reaches the minimum or $10^4$ epochs is exceeded (horizontal dashed line). This plot show that under SGD, the error either starts to increases from the first epoch (filled squares) or keeps decreasing on average for at least $10^4$ epochs (unfilled squares). **B)** The number of epochs, under NP, required to reach the target test error ($\epsilon = 5.0$) under various learning rates and hidden layer widths. **C)** Learning curves of NP and SGD under a small learning rate ($\eta = 10^{-6}$). **D)** Linear dimensionality of the hidden layer activity under NP (orange) and SGD (blue), on the learning trajectories depicted in panel **C** (see Appendix C.1 for details).

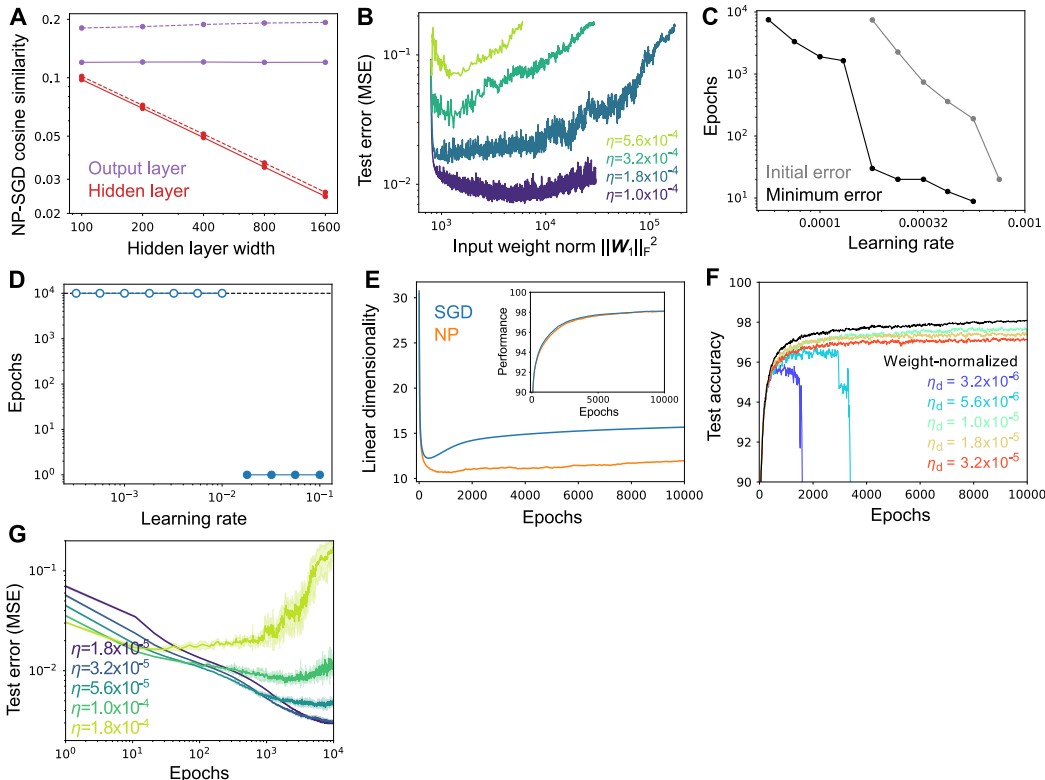

Figure S4: **Nonlinear networks solving MNIST. A)** Cosine similarity between NP and SGD under various hidden layer widths. Solid and dashed lines represent the similarity at two different performance levels (solid: 97.5%, dashed: 90.0%). **B)** Learning dynamics of NP: mean-squared test error versus the input weight norm $\|\mathbf{W}_1\|_F^2$. The purple line ends at $10^4$ epochs; the other lines end when the error exceeded 0.018. **C)** The number of epochs, under NP, until the error starts to increase (black points), and returns to the initial error level (gray points). **D)** The number of epochs, under SGD, until either the error returns to chance level or $10^4$ epochs is exceeded (horizontal dashed line). This plot indicates that under SGD, the error either remains at chance level (filled circles), or stays above it for at least $10^4$ epochs (unfilled circles). **E)** Linear dimensionality of hidden layer activity under NP and SGD at a low learning rate, $\eta = 10^{-5}$. The inset shows the learning trajectories in the same simulations (the orange line is hidden under the blue line). **F)** Comparison of weight normalization (black line; Eq. 108) and weight decay (colored lines) in a network with three hidden layers. Weight decay was implemented by $W \leftarrow (1 - \eta_d)W$ at each update, and the learning rate was fixed at $\eta = 3.2 \times 10^{-5}$. **G)** The same as Fig. 5A, but here we plotted the mean-squared error in a log-log scale.

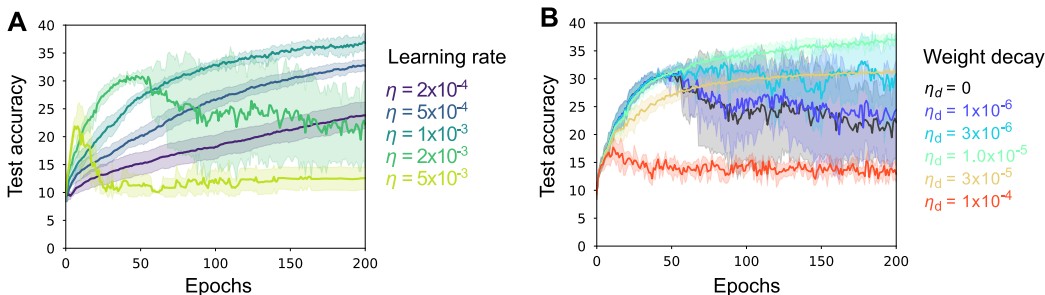

Figure S5: **Convolutional networks solving CIFAR-10. A)** NP learning curves under various learning rates. **B)** Effect of weight decay. Here we fixed the learning rate to $\eta = 2 \times 10^{-3}$. The black line corresponds to the green line in panel **A**. In both panels, shadows represent the standard deviation over 5 random seeds. We used a three layer convolutional neural network each having 32 channels of 3x3 convolution. The output of the last convolutional is projected to fully connected layer with 8192 neurons, then to the output layer. Convolutional network implementation of NP was adopted from `https://anonymous.4open.science/r/nodepert-82E3`.

# Appendix

## A Analysis of nonlinear feedforward neural networks in arbitrary tasks

### A.1 NP is unbiased

As described in the main text, we consider a vanilla deep feedforward network

$$\boldsymbol{x}_k = f_k(\boldsymbol{W}_k \boldsymbol{x}_{k-1}),\ k = 1, 2, ..., K \tag{16}$$

where $\boldsymbol{x}_0$ and $\boldsymbol{x}_K$ are the input and the output respectively, and $f_k(\cdot)$ is an element-wise activation function. Here, the $k$-th layer activity $\boldsymbol{x}_k$ and pre-synaptic weight $\boldsymbol{W}_k$ satisfy $\boldsymbol{x}_k \in R^{L_k}$ and $\boldsymbol{W}_k \in R^{L_k \times L_{k-1}}$, where $L_k$ is the size of the $k$-th layer. Throughout the paper, we used bold italic lower case letters to denote column vectors and bold italic upper case letters to denote matrices. Adding a small Gaussian perturbation $\sigma \boldsymbol{\xi}_k$ to each layer except the input layer, gives us the perturbed activity

$$\widetilde{\boldsymbol{x}}_k = f_k(\boldsymbol{W}_k \widetilde{\boldsymbol{x}}_{k-1} + \sigma \boldsymbol{\xi}_k),\ k = 1, 2, ..., K. \tag{17}$$

Under a loss function $\ell(\boldsymbol{x}_K, \boldsymbol{x}_0)$, the node perturbation update is defined as

$$\delta \boldsymbol{W}_k^{np} = -\frac{\eta}{\sigma} \left( \ell(\widetilde{\boldsymbol{x}}_K, \boldsymbol{x}_0) - \ell(\boldsymbol{x}_K, \boldsymbol{x}_0) \right) \boldsymbol{\xi}_k \boldsymbol{x}_{k-1}^T. \tag{18}$$

In the limit $\sigma \to 0$,

$$\tilde{\boldsymbol{x}}_K = \boldsymbol{x}_K + \sigma \sum_{k=1}^{K} \frac{\partial \boldsymbol{x}_K}{\partial \boldsymbol{h}_k} \boldsymbol{\xi}_k, \tag{19}$$

where

$$\boldsymbol{h}_k \equiv \boldsymbol{W}_k \boldsymbol{x}_{k-1}. \tag{20}$$

Thus, denoting

$$\boldsymbol{g}_k \equiv \frac{\partial \ell}{\partial \boldsymbol{x}_K} \frac{\partial \boldsymbol{x}_K}{\partial \boldsymbol{h}_k}, \tag{21}$$

where $\boldsymbol{g}_k \in R^{L_k}$, in the small $\sigma$ limit the NP update becomes

$$\delta \boldsymbol{W}_k^{np} = -\eta \sum_{l=1}^{K} \boldsymbol{\xi}_k \boldsymbol{\xi}_l^T \boldsymbol{g}_l \boldsymbol{x}_{k-1}^T. \tag{22}$$

As in the main text, taking expectation over $\boldsymbol{\xi}$ gives us

$$\langle \delta \boldsymbol{W}_k^{np} \rangle_{\boldsymbol{\xi}} = -\eta \boldsymbol{g}_k \boldsymbol{x}_{k-1}^T. \tag{23}$$

On the other hand, the SGD update is given by

$$\delta \boldsymbol{W}_k^{sgd} = -\eta \frac{\partial \ell(\boldsymbol{x}_K, \boldsymbol{x}_o)}{\partial \boldsymbol{W}_k} = -\eta \frac{\partial \ell(\boldsymbol{x}_K, \boldsymbol{x}_o)}{\partial \boldsymbol{h}_k} \boldsymbol{x}_{k-1}^T = -\eta \boldsymbol{g}_k \boldsymbol{x}_{k-1}^T. \tag{24}$$

Therefore, NP is unbiased against SGD. Moreover, because SGD with i.i.d. samples is unbiased against the true gradient, NP is also unbiased against it.

## A.2 Cosine angle between SGD and NP updates

By definition,

$$\left\langle \left\| \delta \boldsymbol{W}_k^{sgd} \right\|_F^2 \right\rangle_{\boldsymbol{x}} = \eta^2 \left\langle \text{tr} \left[ \boldsymbol{x}_{k-1} \boldsymbol{g}_k^T \boldsymbol{g}_k \boldsymbol{x}_{k-1}^T \right] \right\rangle_{\boldsymbol{x}} = \eta^2 \left\langle \| \boldsymbol{g}_k \|^2 \, \| \boldsymbol{x}_{k-1} \|^2 \right\rangle_{\boldsymbol{x}}, \tag{25a}$$

$$\left\langle \| \delta \boldsymbol{W}_k^{np} \|_F^2 \right\rangle_{\boldsymbol{x}, \boldsymbol{\xi}} = \eta^2 \sum_{l=1}^K \sum_{m=1}^K \left\langle \text{tr} \left[ \boldsymbol{x}_{k-1} \boldsymbol{g}_l^T \boldsymbol{\xi}_l \boldsymbol{\xi}_k^T \boldsymbol{\xi}_k \boldsymbol{\xi}_m^T \boldsymbol{g}_m \boldsymbol{x}_{k-1}^T \right] \right\rangle_{\boldsymbol{x}, \boldsymbol{\xi}}$$

$$= \eta^2 \sum_l \left\langle \text{tr} \left[ (L_k + 2\delta_{kl}) \boldsymbol{x}_{k-1} \boldsymbol{g}_l^T \boldsymbol{g}_l \boldsymbol{x}_{k-1}^T \right] \right\rangle_{\boldsymbol{x}}$$

$$= \eta^2 \sum_l (L_k + 2\delta_{kl}) \left\langle \| \boldsymbol{g}_l \|^2 \, \| \boldsymbol{x}_{k-1} \|^2 \right\rangle_{\boldsymbol{x}}, \tag{25b}$$

$$\left\langle \text{tr} \left[ \left( \delta \boldsymbol{W}_k^{sgd} \right)^T \delta \boldsymbol{W}_k^{np} \right] \right\rangle_{\boldsymbol{x}, \boldsymbol{\xi}} = \eta^2 \sum_l \left\langle \text{tr} \left[ \boldsymbol{x}_{k-1} \boldsymbol{g}_k^T \boldsymbol{\xi}_k \boldsymbol{\xi}_l^T \boldsymbol{g}_l \boldsymbol{x}_{k-1}^T \right] \right\rangle_{\boldsymbol{x}, \boldsymbol{\xi}} = \eta^2 \left\langle \| \boldsymbol{g}_k \|^2 \, \| \boldsymbol{x}_{k-1} \|^2 \right\rangle_{\boldsymbol{x}}. \tag{25c}$$

In the second equation, we used

$$\begin{aligned}
\langle \boldsymbol{\xi}_l \boldsymbol{\xi}_k^T \boldsymbol{\xi}_k \boldsymbol{\xi}_m^T \rangle &= \delta_{lm} \langle \boldsymbol{\xi}_l \boldsymbol{\xi}_k^T \boldsymbol{\xi}_k \boldsymbol{\xi}_l^T \rangle \\
&= \delta_{lm} ( [1 - \delta_{kl}] \langle \boldsymbol{\xi}_k^T \boldsymbol{\xi}_k \rangle \langle \boldsymbol{\xi}_l \boldsymbol{\xi}_l^T \rangle + \delta_{kl} \langle \boldsymbol{\xi}_k \boldsymbol{\xi}_k^T \boldsymbol{\xi}_k \boldsymbol{\xi}_k^T \rangle ) \\
&= \delta_{lm} ( [1 - \delta_{kl}] L_k \boldsymbol{I}_k + \delta_{kl} [L_k + 2] \boldsymbol{I}_k ) \\
&= \delta_{lm} ( L_k + 2\delta_{kl} ) \boldsymbol{I}_k, 
\end{aligned} \tag{26}$$

where $\boldsymbol{I}_k$ is the size $L_k$ identity matrix. Note that, the $(\mu, \nu)$-th element of matrix $\langle \boldsymbol{\xi}_k \boldsymbol{\xi}_k^T \boldsymbol{\xi}_k \boldsymbol{\xi}_k^T \rangle$ is given as

$$\left[ \langle \boldsymbol{\xi}_k \boldsymbol{\xi}_k^T \boldsymbol{\xi}_k \boldsymbol{\xi}_k^T \rangle \right]_{\mu\nu} = \delta_{\mu\nu} \sum_{\rho=1}^{L_k} \left( [1 - \delta_{\mu\rho}] \langle (\xi_\mu^k)^2 \rangle \langle (\xi_\rho^k)^2 \rangle + \delta_{\mu\rho} \langle (\xi_\mu^k)^4 \rangle \right) = \delta_{\mu\nu} (L_k + 2). \tag{27}$$

The cosine of the angle between SGD and NP update is thus estimated as

$$\left\langle \frac{\text{tr} \left[ \left( \delta \boldsymbol{W}_k^{sgd} \right)^T \delta \boldsymbol{W}_k^{np} \right]}{\sqrt{\left\| \delta \boldsymbol{W}_k^{sgd} \right\|_F^2 \| \delta \boldsymbol{W}_k^{np} \|_F^2}} \right\rangle_{\boldsymbol{x}, \boldsymbol{\xi}} \approx \frac{\left\langle \text{tr} \left[ \left( \delta \boldsymbol{W}_k^{sgd} \right)^T \delta \boldsymbol{W}_k^{np} \right] \right\rangle_{\boldsymbol{x}}}{\sqrt{\left\langle \left\| \delta \boldsymbol{W}_k^{sgd} \right\|_F^2 \right\rangle \left\langle \| \delta \boldsymbol{W}_k^{np} \|_F^2 \right\rangle_{\boldsymbol{x}}}}$$

$$= \sqrt{\frac{\left\langle \| \boldsymbol{g}_k \|^2 \, \| \boldsymbol{x}_{k-1} \|^2 \right\rangle_{\boldsymbol{x}}}{\sum_{l=1}^K (L_k + 2\delta_{kl}) \left\langle \| \boldsymbol{g}_l \|^2 \, \| \boldsymbol{x}_{k-1} \|^2 \right\rangle_{\boldsymbol{x}}}}. \tag{28}$$

This is a good approximation if the higher-order terms stem from the coupling between the numerator and the denominator are negligible.

## A.3 Covariance of SGD and NP updates

Another way to characterize noise in SGD and NP updates is to calculate the covariance of the weight change under each update rule. The true gradient is written as

$$\delta \boldsymbol{W}_k^{gd} = -\eta \left\langle \boldsymbol{g}_k \boldsymbol{x}_{k-1}^T \right\rangle_{\boldsymbol{x}}. \tag{29}$$

Therefore, the noise covariance of SGD updates between the $k$-th and $l$-th layers is given by

$$C_{kl}^{sgd} = \left\langle \left( \boldsymbol{g}_k \boldsymbol{x}_{k-1}^T - \left\langle \boldsymbol{g}_k \boldsymbol{x}_{k-1}^T \right\rangle_{\boldsymbol{x}} \right) \otimes \left( \boldsymbol{g}_l \boldsymbol{x}_{l-1}^T - \left\langle \boldsymbol{g}_l \boldsymbol{x}_{l-1}^T \right\rangle_{\boldsymbol{x}} \right) \right\rangle_{\boldsymbol{x}}. \tag{30}$$

where, as in the main text, $\otimes$ represents a tensor product (see comments following Eq. 7). Here, we write the covariance matrix as a fourth-order tensor for clarity. If the true gradient is smaller than SGD variance [43], this simplifies to

$$C_{kl}^{sgd} \approx \left\langle \boldsymbol{g}_k \boldsymbol{x}_{k-1}^T \otimes \boldsymbol{g}_l \boldsymbol{x}_{l-1}^T \right\rangle_{\boldsymbol{x}}. \tag{31}$$

Similarly, noticing that

$$\delta \boldsymbol{W}_k^{np} - \delta \boldsymbol{W}_k^{gd} = \delta \boldsymbol{W}_k^{np} - \delta \boldsymbol{W}_k^{sgd} + \delta \boldsymbol{W}_k^{sgd} - \delta \boldsymbol{W}_k^{gd}$$

$$= -\eta \left( \sum_l \boldsymbol{\xi}_k \boldsymbol{\xi}_l^T \boldsymbol{g}_l - \boldsymbol{g}_k \right) \boldsymbol{x}_{k-1}^T - \eta \left( \boldsymbol{g}_k \boldsymbol{x}_{k-1}^T - \langle \boldsymbol{g}_k \boldsymbol{x}_{k-1}^T \rangle_{\boldsymbol{x}} \right), \quad (32)$$

the covariance of NP update is derived as

$$C_{kl}^{np} = \left\langle \left[ \sum_m \boldsymbol{\xi}_k \boldsymbol{\xi}_m^T \boldsymbol{g}_m - \boldsymbol{g}_k \right] \boldsymbol{x}_{k-1}^T \otimes \left[ \sum_n \boldsymbol{\xi}_l \boldsymbol{\xi}_n^T \boldsymbol{g}_n - \boldsymbol{g}_l \right] \boldsymbol{x}_{l-1}^T \right\rangle_{\boldsymbol{x},\boldsymbol{\xi}} + C_{kl}^{sgd}. \quad (33)$$

Focusing on the $\boldsymbol{\xi}$ dependent term, we have

$$\left\langle \left[ \sum_m \boldsymbol{\xi}_k \boldsymbol{\xi}_m^T \boldsymbol{g}_m - \boldsymbol{g}_k \right] \left[ \sum_n \boldsymbol{\xi}_l \boldsymbol{\xi}_n^T \boldsymbol{g}_n - \boldsymbol{g}_l \right]^T \right\rangle_{\boldsymbol{\xi}} = \sum_m \sum_n \left\langle \boldsymbol{\xi}_k \boldsymbol{\xi}_m^T \boldsymbol{g}_m \boldsymbol{g}_n^T \boldsymbol{\xi}_n \boldsymbol{\xi}_l^T \right\rangle_{\boldsymbol{\xi}} - \boldsymbol{g}_k \boldsymbol{g}_l^T$$

$$= \boldsymbol{g}_k \boldsymbol{g}_l^T + \delta_{kl} \sum_m \|\boldsymbol{g}_m\|^2 \, \boldsymbol{I}_k. \quad (34)$$

In the last line, we used

$$\sum_m \sum_n \left[ \langle \boldsymbol{\xi}_k \boldsymbol{\xi}_m^T \boldsymbol{g}_m \boldsymbol{g}_n^T \boldsymbol{\xi}_n \boldsymbol{\xi}_l^T \rangle_{\boldsymbol{\xi}} \right]_{\mu\nu} = \sum_m \sum_n \sum_\rho \sum_\lambda \langle \xi_\mu^k \xi_\rho^m g_\rho^m g_\lambda^n \xi_\lambda^n \xi_\nu^l \rangle_{\boldsymbol{\xi}}$$

$$= \sum_m \sum_n \sum_\rho \sum_\lambda g_\rho^m g_\lambda^n \left( \delta_{\mu\rho}^{km} \delta_{\lambda\nu}^{nl} + \delta_{\mu\lambda}^{kn} \delta_{\rho\nu}^{ml} + \delta_{\mu\nu}^{kl} \delta_{\rho\lambda}^{mn} \right)$$

$$= 2 g_\mu^k g_\nu^l + \delta_{\mu\nu}^{kl} \sum_m \sum_\rho (g_\rho^m)^2. \quad (35)$$

Thus, taking a transpose between the second and the third dimension, the covariance tensor is written as

$$C_{kl}^{np} = \left\langle \left( \delta_{kl} \sum_m \|\boldsymbol{g}_m\|^2 \, \boldsymbol{I}_k + \boldsymbol{g}_k \boldsymbol{g}_l^T \right) \otimes \boldsymbol{x}_{k-1} \boldsymbol{x}_{l-1}^T \right\rangle_{\boldsymbol{x}} + C_{kl}^{sgd}$$

$$= 2 C_{kl}^{sgd} + \left[ \langle \boldsymbol{g}_k \boldsymbol{x}_{k-1} \rangle \otimes \langle \boldsymbol{g}_l \boldsymbol{x}_{l-1}^T \rangle_{\boldsymbol{x}} \right]_{2\leftrightarrow 3} + \delta_{kl} \left\langle \sum_m \|\boldsymbol{g}_m\|^2 \, \boldsymbol{I}_k \otimes \boldsymbol{x}_{k-1} \boldsymbol{x}_{k-1}^T \right\rangle_{\boldsymbol{x}} \quad (36a)$$

$$\approx 2 C_{kl}^{sgd} + \delta_{kl} \left\langle \sum_m \|\boldsymbol{g}_m\|^2 \, \boldsymbol{I}_k \otimes \boldsymbol{x}_{k-1} \boldsymbol{x}_{k-1}^T \right\rangle_{\boldsymbol{x}}, \quad (36b)$$

where $[\cdot]_{2\leftrightarrow 3}$ in the second line denotes a transpose between the second and the third dimension. In the last line, we assumed that $\langle \boldsymbol{g}_k \boldsymbol{x}_{k-1}^T \rangle \approx 0$, as in Eq. 31. Notably, if the activity at each layer is whitened as $\boldsymbol{x}_{k-1} \boldsymbol{x}_{k-1}^T \approx \boldsymbol{I}_{k-1}$, the second term in the last line is diagonal, meaning that NP predominantly increases auto-covariance of the update at each synapse compared to SGD while keeping the noise correlation between the updates at different synapses comparable to SGD.

## A.4 Noise in over-parameterized networks

In convex optimization, noise in the gradients becomes more harmful as the dimensionality of the system goes up [33, 29]. However, in an over-parameterized neural network, most directions in parameter space are irrelevant to learning [44, 45], so noise might not be all that harmful after all. To further study how noise affects the change in loss function, note that both SGD and NP are written as

$$\delta \boldsymbol{w} = -\eta (\nabla L + \boldsymbol{z}), \quad (37)$$

where the noise term is approximately Gaussian ($\boldsymbol{z} \sim N(0, \boldsymbol{C})$). Under the second order approximation, the change in the loss function is given as

$$\delta L \equiv [\boldsymbol{w} + \delta \boldsymbol{w}] - L[\boldsymbol{w}] \approx \nabla L[\boldsymbol{w}] \cdot \delta \boldsymbol{w} + \frac{1}{2} \delta \boldsymbol{w}^T \nabla^2 L[\boldsymbol{w}] \delta \boldsymbol{w}. \quad (38)$$

Taking an expectation over the noise,

$$\langle \delta L \rangle_z \approx -\eta \, \|\nabla L[\boldsymbol{w}]\|^2 + \frac{\eta^2}{2} (\nabla L[\boldsymbol{w}])^T \nabla^2 L[\boldsymbol{w}] \nabla L[\boldsymbol{w}] + \frac{\eta^2}{2} \text{tr} \left[ \nabla^2 L \cdot \boldsymbol{C} \right] . \tag{39}$$

Thus, the noise affects the update only through the inner product with the Hessian $\nabla^2 L$. This means that if the noise is in the direction of a major positive eigenvector of the Hessian, the noise is harmful, but otherwise, noise is likely to be benign.

# B  Analysis of one hidden layered linear networks in the student teacher setting

## B.1  Problem setting

Here we consider a student-teacher setting. The student network is given by

$$\boldsymbol{y} = \boldsymbol{W}_2 \boldsymbol{W}_1 \boldsymbol{x}, \tag{40}$$

where $\boldsymbol{W}_2$ and $\boldsymbol{W}_1$ are matrices of size $L_y \times L_h$ and $L_h \times L_x$, respectively. For the teacher network, we add up noise as,

$$\boldsymbol{y}^* = \boldsymbol{A} \boldsymbol{x} + \sigma_t \boldsymbol{z}, \tag{41}$$

where $\boldsymbol{z}$ is a zero mean independent Gaussian random variable ($\boldsymbol{z} \sim N(0, \boldsymbol{I})$) and $\sigma_t$ is the noise amplitude. The teacher noise is introduced to replicate the mismatch between the student and the teacher models that typically exists in a general setting. For analytical tractability, we sampled the input, $\boldsymbol{x}$, from an independent Gaussian distribution ($\boldsymbol{x} \sim N(0, \boldsymbol{I})$).

We use the MSE error as the loss function,

$$\ell(\boldsymbol{x}; \boldsymbol{W}) = \frac{1}{2} \|\boldsymbol{y} - \boldsymbol{y}^*\|^2 , \tag{42}$$

and denote the signed weight error as

$$\boldsymbol{E} \equiv \boldsymbol{W}_2 \boldsymbol{W}_1 - \boldsymbol{A} . \tag{43}$$

To track the learning dynamics, we introduce the following order-parameters,

$$\epsilon \equiv \frac{1}{L_x L_y} \text{tr}[\boldsymbol{E}^T \boldsymbol{E}], \quad \phi \equiv \frac{-1}{L_x L_y} \text{tr}[(\boldsymbol{W}_2 \boldsymbol{W}_1)^T \boldsymbol{E}], \tag{44a}$$

$$\alpha \equiv \frac{1}{L_x} \text{tr}[\boldsymbol{W}_1^T \boldsymbol{W}_1], \quad \beta \equiv \frac{1}{L_y} \text{tr}[\boldsymbol{W}_2 \boldsymbol{W}_2^T]. \tag{44b}$$

Here $\epsilon$ is the error, $\phi$ is the projection of the student model in the direction of the error, and $\alpha$ and $\beta$ represent the weight norm of the input weight and the output weight, respectively. Below, we study how these four order parameters evolve under NP and SGD in the limit where both the input and output dimensions are large, $L_x \gg 1$ and $L_y \gg 1$, and the hidden layer dimension is large compared to both of them, $L_h \gg L_x$ and $L_h \gg L_y$. In the simulations, we generated $\boldsymbol{A}$ randomly with $A_{ij} \sim N(0, 2/(L_x + L_y))$, and we used this condition in Appendix B.3. However, the analytical results depicted in Appendices B.2 and B.4 don't rely on the random teacher assumption.

## B.2  Mean-field dynamics under NP

When we add small perturbations to the both intermediate and the output layers, the perturbed output becomes

$$\tilde{\boldsymbol{y}} = \boldsymbol{W}_2 \left( \boldsymbol{W}_1 \boldsymbol{x} + \sigma \boldsymbol{\xi}_1 \right) + \sigma \boldsymbol{\xi}_2. \tag{45}$$

At the small perturbation limit ($\sigma \to 0$), the difference between the perturbed and the original loss is

$$\ell(\tilde{\boldsymbol{y}}, \boldsymbol{x}) - \ell(\boldsymbol{y}, \boldsymbol{x}) \approx \sigma \left( \boldsymbol{\xi}_2^T + \boldsymbol{\xi}_1^T \boldsymbol{W}_2^T \right) (\boldsymbol{E} \boldsymbol{x} + \sigma_t \boldsymbol{z}). \tag{46}$$

Note that here we flipped the sign of $\boldsymbol{z}$, because $\boldsymbol{z}$ is a zero-mean random Gaussian variable. Thus, NP learning rule is given by

$$\delta \boldsymbol{W}_1 = -\eta \left( \boldsymbol{\xi}_1 \boldsymbol{\xi}_1^T \boldsymbol{W}_2^T + \boldsymbol{\xi}_1 \boldsymbol{\xi}_2^T \right) \left( \boldsymbol{E} \boldsymbol{x} \boldsymbol{x}^T + \sigma_t \boldsymbol{z} \boldsymbol{x}^T \right), \tag{47}$$

$$\delta \boldsymbol{W}_2 = -\eta \left( \boldsymbol{\xi}_2 \boldsymbol{\xi}_2^T + \boldsymbol{\xi}_2 \boldsymbol{\xi}_1^T \boldsymbol{W}_2^T \right) \left( \boldsymbol{E} \boldsymbol{x} \boldsymbol{x}^T + \sigma_t \boldsymbol{z} \boldsymbol{x}^T \right) \boldsymbol{W}_1^T. \tag{48}$$

Below, we investigate how the order parameters defined in Eq. 44 change during learning. We work to second order in the learning rate, $\eta$, which corresponds to second order in $\delta \boldsymbol{W}_1$ and $\delta \boldsymbol{W}_2$. First, the change in the input weight norm is estimated as

$$\langle \delta \alpha \rangle_{\boldsymbol{x}, \boldsymbol{\xi}, \boldsymbol{z}} \equiv \left\langle \frac{1}{L_x} \mathrm{tr} \left[ (\boldsymbol{W}_1 + \delta \boldsymbol{W}_1)^T (\boldsymbol{W}_1 + \delta \boldsymbol{W}_1) \right] - \frac{1}{L_x} \mathrm{tr} \left[ \boldsymbol{W}_1^T \boldsymbol{W}_1 \right] \right\rangle_{\boldsymbol{x}, \boldsymbol{\xi}, \boldsymbol{z}}$$

$$= \frac{2}{L_x} \left\langle \mathrm{tr}[\boldsymbol{W}_1^T \delta \boldsymbol{W}_1] \right\rangle_{\boldsymbol{x}, \boldsymbol{\xi}, \boldsymbol{z}} + \frac{1}{L_x} \left\langle \mathrm{tr}[\delta \boldsymbol{W}_1^T \delta \boldsymbol{W}_1] \right\rangle_{\boldsymbol{x}, \boldsymbol{\xi}, \boldsymbol{z}}. \tag{49}$$

The first term simplifies to

$$\frac{2}{L_x} \left\langle \mathrm{tr}[\boldsymbol{W}_1^T \delta \boldsymbol{W}_1] \right\rangle = -\frac{2\eta}{L_x} \mathrm{tr}[\boldsymbol{W}_1^T \boldsymbol{W}_2^T \boldsymbol{E}] = 2\eta L_y \phi. \tag{50}$$

The second equality follows from the definition of $\phi$ (Eq. 44). The noise term, on the other hand, is somewhat more complicated,

$$\frac{1}{L_x} \left\langle \mathrm{tr}[\delta \boldsymbol{W}_1^T \delta \boldsymbol{W}_1] \right\rangle = \frac{\eta^2}{L_x} \left\langle \mathrm{tr}[(\boldsymbol{x}\boldsymbol{x}^T \boldsymbol{E}^T + \sigma_t \boldsymbol{x}\boldsymbol{z}^T)(\boldsymbol{W}_2 \boldsymbol{\xi}_1 \boldsymbol{\xi}_1^T + \boldsymbol{\xi}_2 \boldsymbol{\xi}_1^T)(\boldsymbol{\xi}_1 \boldsymbol{\xi}_1^T \boldsymbol{W}_2^T + \boldsymbol{\xi}_1 \boldsymbol{\xi}_2^T)(\boldsymbol{E} \boldsymbol{x}\boldsymbol{x}^T + \sigma_t \boldsymbol{z}\boldsymbol{x}^T)] \right\rangle$$

$$= \frac{\eta^2}{L_x} \left\langle \mathrm{tr}[(\boldsymbol{W}_2 \boldsymbol{\xi}_1 \boldsymbol{\xi}_1^T \boldsymbol{\xi}_1 \boldsymbol{\xi}_1^T \boldsymbol{W}_2^T + \boldsymbol{\xi}_2 \boldsymbol{\xi}_1^T \boldsymbol{\xi}_1 \boldsymbol{\xi}_2^T)(\boldsymbol{E} \boldsymbol{x}\boldsymbol{x}^T \boldsymbol{x}\boldsymbol{x}^T \boldsymbol{E}^T + \sigma_t^2 \boldsymbol{z}\boldsymbol{x}^T \boldsymbol{x}\boldsymbol{z}^T)] \right\rangle$$

$$\approx \frac{\eta^2}{L_x} \mathrm{tr}[L_h (\boldsymbol{W}_2 \boldsymbol{W}_2^T + \boldsymbol{I}_y) L_x (\boldsymbol{E}\boldsymbol{E}^T + \sigma_t^2 \boldsymbol{I}_y)].$$

$$= \eta^2 L_h \left( L_y [1 + \beta] \sigma_t^2 + L_x L_y \epsilon + \mathrm{tr}[\boldsymbol{W}_2 \boldsymbol{W}_2^T \boldsymbol{E}\boldsymbol{E}^T] \right). \tag{51}$$

Note that, by taking the expectation over $\boldsymbol{\xi}$, $\boldsymbol{z}$, and $\boldsymbol{x}$, the second line of the equation above is rewritten as $\mathrm{tr} \left[ \left( [L_h + 2] \boldsymbol{W}_2 \boldsymbol{W}_2^T + L_h \boldsymbol{I}_y \right) \left( [L_x + 2] \boldsymbol{E}\boldsymbol{E}^T + \sigma_t^2 L_x \boldsymbol{I}_y \right) \right]$. Approximating $L_h + 2$ with $L_h$ and $L_x + 2$ with $L_x$ using the assumption $L_x, L_h \gg 1$, we get the third line of the equation. Because "+2" terms dropped in the approximation come from higher-order moments of Gaussian random variables, we call this approximation dropping the higher-order terms.

To express the average change in $\alpha$ in terms of $\beta$, $\epsilon$, and $\phi$, we need to express $\mathrm{tr}[\boldsymbol{W}_2 \boldsymbol{W}_2^T \boldsymbol{E}\boldsymbol{E}^T]$ in terms of the order parameters. To do that, we introduce the following, rather severe, approximation (see Appendix B.5 for the details),

$$\mathrm{tr}[\boldsymbol{W}_2 \boldsymbol{W}_2^T \boldsymbol{E}\boldsymbol{E}^T] \approx \frac{1}{L_y} \mathrm{tr}[\boldsymbol{W}_2 \boldsymbol{W}_2^T] \mathrm{tr}[\boldsymbol{E}\boldsymbol{E}^T] = L_x L_y \beta \epsilon \tag{52}$$

In the large $L_h$ limit, this approximation holds early in learning if $\boldsymbol{W}_1$ and $\boldsymbol{W}_2$ are initialized as zero-mean random Gaussian matrices. Moreover, if the elements of the weight updates $\delta \boldsymbol{W}_1$ and $\delta \boldsymbol{W}_2$ are zero-mean Gaussian random variables, the approximation is still valid (Appendix B.5). Because the weight update under NP is dominated by noise (Appendix A.2), and the noise covariance matrix is nearly diagonal (Eq. 36a), this assumption approximately holds under NP. By contrast, its applicability to SGD is limited, especially under a mini-batch formulation. Even under NP, after many weight updates, the approximation ceases to hold.

Using Eq. 52, the average change in the input weight norm is estimated as

$$\langle \delta \alpha \rangle \approx 2\eta L_y \phi + \eta^2 L_h L_y (1 + \beta)(L_x \epsilon + \sigma_t^2). \tag{53}$$

This result suggests that the input weight norm expands rapidly when the hidden layer size $L_h$ is large. Notably, this $L_h$ dependence is absent under SGD (see Eq. 88).

Similarly, the average change in $\beta$ after one update is written as,

$$\langle \delta \beta \rangle_{\boldsymbol{x}, \boldsymbol{\xi}, \boldsymbol{z}} \equiv \left\langle \frac{1}{L_y} \mathrm{tr} \left[ (\boldsymbol{W}_2 + \delta \boldsymbol{W}_2)(\boldsymbol{W}_2 + \delta \boldsymbol{W}_2)^T \right] - \frac{1}{L_y} \mathrm{tr} \left[ \boldsymbol{W}_2 \boldsymbol{W}_2^T \right] \right\rangle_{\boldsymbol{x}, \boldsymbol{\xi}, \boldsymbol{z}}$$

$$= \frac{2}{L_y} \left\langle \mathrm{tr}[\boldsymbol{W}_2 \delta \boldsymbol{W}_2^T] \right\rangle_{\boldsymbol{x}, \boldsymbol{\xi}, \boldsymbol{z}} + \frac{1}{L_y} \left\langle \mathrm{tr}[\delta \boldsymbol{W}_2 \delta \boldsymbol{W}_2^T] \right\rangle_{\boldsymbol{x}, \boldsymbol{\xi}, \boldsymbol{z}}.$$

As is straightforward to show, the signal term given by $(2/L_y)\left\langle \mathrm{tr}[\boldsymbol{W}_2\delta\boldsymbol{W}_2^T]\right\rangle = 2\eta L_x\phi$, while the noise term of $\langle\delta\beta\rangle$ is given by

$$\frac{1}{L_y}\left\langle \mathrm{tr}[\delta\boldsymbol{W}_2^T\delta\boldsymbol{W}_2]\right\rangle$$

$$= \frac{\eta^2}{L_y}\left\langle \mathrm{tr}[\boldsymbol{W}_1(\boldsymbol{x}\boldsymbol{x}^T\boldsymbol{E}^T + \sigma_t\boldsymbol{x}\boldsymbol{z}^T)(\boldsymbol{\xi}_2\boldsymbol{\xi}_2^T + \boldsymbol{W}_2\boldsymbol{\xi}_1\boldsymbol{\xi}_2^T)(\boldsymbol{\xi}_2\boldsymbol{\xi}_2^T + \boldsymbol{\xi}_2\boldsymbol{\xi}_1^T\boldsymbol{W}_2^T)(\boldsymbol{E}\boldsymbol{x}\boldsymbol{x}^T + \sigma_t\boldsymbol{z}\boldsymbol{x}^T)\boldsymbol{W}_1^T]\right\rangle$$

$$= \frac{\eta^2}{L_y}\left\langle \mathrm{tr}[(\boldsymbol{\xi}_2\boldsymbol{\xi}_2^T\boldsymbol{\xi}_2\boldsymbol{\xi}_2^T + \boldsymbol{W}_2\boldsymbol{\xi}_1\boldsymbol{\xi}_2^T\boldsymbol{\xi}_2\boldsymbol{\xi}_1^T\boldsymbol{W}_2^T)(\boldsymbol{E}\boldsymbol{x}\boldsymbol{x}^T\boldsymbol{W}_1^T\boldsymbol{W}_1\boldsymbol{x}\boldsymbol{x}^T\boldsymbol{E}^T + \sigma_t^2\boldsymbol{z}\boldsymbol{x}^T\boldsymbol{W}_1^T\boldsymbol{W}_1\boldsymbol{x}\boldsymbol{z}^T)]\right\rangle$$

$$\approx \frac{\eta^2}{L_y}\left(\mathrm{tr}[L_y(\boldsymbol{I}_y + \boldsymbol{W}_2\boldsymbol{W}_2^T)(\boldsymbol{E}\boldsymbol{E}^T + \sigma_t^2\boldsymbol{I}_y)]\mathrm{tr}[\boldsymbol{W}_1^T\boldsymbol{W}_1]\right)$$

$$\approx \eta^2 L_x L_y\alpha(1+\beta)(L_x\epsilon + \sigma_t^2). \tag{54}$$

In the last line, we again used Eq. 52. Thus, the average change in $\beta$ is

$$\langle\delta\beta\rangle \approx 2\eta L_x\phi + \eta^2 L_x L_y\alpha(1+\beta)(L_x\epsilon + \sigma_t^2). \tag{55}$$

The dynamics of the error $\epsilon$ is, up to the second order term with respect to $\eta$, given by

$$\langle\delta\epsilon\rangle = \frac{2}{L_x L_y}\left\langle \mathrm{tr}[\boldsymbol{E}^T(\boldsymbol{W}_2\delta\boldsymbol{W}_1 + \delta\boldsymbol{W}_2\boldsymbol{W}_1)]\right\rangle + \frac{2}{L_x L_y}\left\langle \mathrm{tr}[\boldsymbol{E}^T\delta\boldsymbol{W}_2\delta\boldsymbol{W}_1]\right\rangle$$

$$+ \frac{1}{L_x L_y}\left\langle \mathrm{tr}[(\boldsymbol{W}_2\delta\boldsymbol{W}_1 + \delta\boldsymbol{W}_2\boldsymbol{W}_1)^T(\boldsymbol{W}_2\delta\boldsymbol{W}_1 + \delta\boldsymbol{W}_2\boldsymbol{W}_1)]\right\rangle. \tag{56}$$

Using an approximation parallel to Eq. 52,

$$\mathrm{tr}[\boldsymbol{W}_1^T\boldsymbol{W}_1\boldsymbol{E}^T\boldsymbol{E}] \approx \frac{1}{L_x}\mathrm{tr}[\boldsymbol{W}_1^T\boldsymbol{W}_1]\mathrm{tr}[\boldsymbol{E}^T\boldsymbol{E}] = L_x L_y\alpha\epsilon, \tag{57}$$

the signal term is given as

$$\frac{2}{L_x L_y}\left\langle \mathrm{tr}[\boldsymbol{E}^T(\boldsymbol{W}_2\delta\boldsymbol{W}_1 + \delta\boldsymbol{W}_2\boldsymbol{W}_1)]\right\rangle \approx -2\eta(\alpha + \beta)\epsilon. \tag{58}$$

The first noise term becomes

$$\frac{2}{L_x L_y}\left\langle \mathrm{tr}[\boldsymbol{E}^T\delta\boldsymbol{W}_2\delta\boldsymbol{W}_1]\right\rangle$$

$$= \frac{2\eta^2}{L_x L_y}\left\langle \mathrm{tr}[\boldsymbol{E}^T(\boldsymbol{\xi}_2\boldsymbol{\xi}_2^T + \boldsymbol{\xi}_2\boldsymbol{\xi}_1^T\boldsymbol{W}_2^T)(\boldsymbol{E}\boldsymbol{x}\boldsymbol{x}^T + \sigma_t\boldsymbol{z}\boldsymbol{x}^T)\boldsymbol{W}_1^T(\boldsymbol{\xi}_1\boldsymbol{\xi}_1^T\boldsymbol{W}_2^T + \boldsymbol{\xi}_1\boldsymbol{\xi}_2^T)(\boldsymbol{E}\boldsymbol{x}\boldsymbol{x}^T + \sigma_t\boldsymbol{z}\boldsymbol{x}^T)]\right\rangle$$

$$\approx \frac{2\eta^2}{L_x L_y}\left\langle \mathrm{tr}[\boldsymbol{E}^T(\boldsymbol{E}\boldsymbol{x}\boldsymbol{x}^T + \sigma_t\boldsymbol{z}\boldsymbol{x}^T)\boldsymbol{W}_1^T\boldsymbol{W}_2^T(\boldsymbol{E}\boldsymbol{x}\boldsymbol{x}^T + \sigma_t\boldsymbol{z}\boldsymbol{x}^T)]\right\rangle$$

$$+ \left\langle \mathrm{tr}[\boldsymbol{W}_2^T(\boldsymbol{E}\boldsymbol{x}\boldsymbol{x}^T + \sigma_t\boldsymbol{z}\boldsymbol{x}^T)\boldsymbol{W}_1^T]\mathrm{tr}[(\boldsymbol{E}\boldsymbol{x}\boldsymbol{x}^T + \sigma_t\boldsymbol{z}\boldsymbol{x}^T)\boldsymbol{E}^T]\right\rangle$$

$$\approx \frac{4\eta^2}{L_x L_y}\left(\mathrm{tr}[\boldsymbol{E}^T\boldsymbol{E}]\mathrm{tr}[\boldsymbol{W}_1^T\boldsymbol{W}_2^T\boldsymbol{E}] + \sigma_t^2\mathrm{tr}[\boldsymbol{W}_1^T\boldsymbol{W}_2^T\boldsymbol{E}]\right) \approx -4\eta^2\phi(L_x L_y\epsilon + \sigma_t^2). \tag{59}$$

By dropping higher-order terms using $L_h, L_x, L_y \gg 1$, and then applying the approximations Eqs. 52 and 57, the rest of noise terms are calculated as

$$\frac{1}{L_x L_y}\left\langle \mathrm{tr}[\boldsymbol{W}_1^T\delta\boldsymbol{W}_2^T\delta\boldsymbol{W}_2\boldsymbol{W}_1]\right\rangle \approx \frac{\eta^2}{L_x L_y}\mathrm{tr}[L_y(\boldsymbol{I}_y + \boldsymbol{W}_2\boldsymbol{W}_2^T)(\boldsymbol{E}\boldsymbol{E}^T + \sigma_t^2\boldsymbol{I}_y)]\mathrm{tr}[\boldsymbol{W}_1^T\boldsymbol{W}_1\boldsymbol{W}_1^T\boldsymbol{W}_1]$$

$$\approx \eta^2 L_y\alpha_2(1+\beta)(L_x\epsilon + \sigma_t^2), \tag{60a}$$

$$\frac{1}{L_x L_y}\left\langle \mathrm{tr}[\delta\boldsymbol{W}_1^T\boldsymbol{W}_2^T\boldsymbol{W}_2\delta\boldsymbol{W}_1]\right\rangle \approx \frac{\eta^2}{L_x L_y}\mathrm{tr}[\boldsymbol{W}_2^T\boldsymbol{W}_2]\mathrm{tr}[(\boldsymbol{W}_2\boldsymbol{W}_2^T + \boldsymbol{I}_y)L_x(\boldsymbol{E}\boldsymbol{E}^T + \sigma_t^2\boldsymbol{I}_y)]$$

$$\approx \eta^2 L_y\beta(1+\beta)(L_x\epsilon + \sigma_t^2), \tag{60b}$$

$$\frac{1}{L_x L_y}\left\langle \mathrm{tr}[\delta\boldsymbol{W}_1^T\boldsymbol{W}_2^T\delta\boldsymbol{W}_2\boldsymbol{W}_1]\right\rangle \approx \frac{\eta^2}{L_x L_y}\mathrm{tr}[\boldsymbol{W}_1^T\boldsymbol{W}_1]\mathrm{tr}[2\boldsymbol{W}_2\boldsymbol{W}_2^T(\boldsymbol{E}\boldsymbol{E}^T + \sigma_t^2\boldsymbol{I}_y)]$$

$$\approx 2\eta^2\alpha\beta(L_x\epsilon + \sigma_t^2). \tag{60c}$$

At the top equation, we defined $\alpha_2$ by

$$\alpha_2 \equiv \frac{1}{L_x}\mathrm{tr}[\boldsymbol{W}_1\boldsymbol{W}_1^T\boldsymbol{W}_1\boldsymbol{W}_1^T]. \tag{61}$$

The dynamics of $\alpha_2$ is written down as

$$\langle\delta\alpha_2\rangle \approx 4\eta L_y\alpha\phi + 2\eta^2(L_h + L_x)L_y(L_x\epsilon + \sigma_t^2)\alpha(1 + \beta) + 2\eta^2 L_y(L_x[L_y\phi^2 + \alpha\epsilon] + \sigma_t^2[\alpha + \psi]). \tag{62}$$

However, we approximate $\alpha_2$ by $\alpha_2 \approx \alpha^2$ below for simplicity. Summing up the noise terms calculated above, the average change in the error follows

$$\langle\delta\epsilon\rangle = -2\eta(\alpha + \beta)\epsilon - 4\eta^2\phi(L_xL_y\epsilon + \sigma_t^2) + \eta^2\left(L_y[1 + \beta][\alpha^2 + \beta] + 4\alpha\beta\right)(L_x\epsilon + \sigma_t^2). \tag{63}$$

Thus, the error dynamics doesn't directly depend on the hidden layer size $L_h$. However, the noise term scales with the weight norms $\alpha$, which increases rapidly with time under a large $L_h$ (see Eq. 53).

Similarly, the mean dynamics of $\phi$ follows

$$\begin{aligned}
\langle\delta\phi\rangle_{\boldsymbol{x},\boldsymbol{\xi},\boldsymbol{z}} = & \frac{-1}{L_xL_y}\left\langle\mathrm{tr}\left[\left(\boldsymbol{E}^T + \boldsymbol{W}_1^T\boldsymbol{W}_2^T\right)\left(\delta\boldsymbol{W}_2\boldsymbol{W}_1 + \boldsymbol{W}_2\delta\boldsymbol{W}_1\right)\right]\right\rangle \\
& - \frac{1}{L_xL_y}\left\langle\mathrm{tr}\left[\left(\boldsymbol{E}^T + \boldsymbol{W}_1^T\boldsymbol{W}_2^T\right)\delta\boldsymbol{W}_2\delta\boldsymbol{W}_1\right]\right\rangle \\
& - \frac{1}{L_xL_y}\left\langle\mathrm{tr}\left[\left(\delta\boldsymbol{W}_2\boldsymbol{W}_1 + \boldsymbol{W}_2\delta\boldsymbol{W}_1\right)^T\left(\delta\boldsymbol{W}_2\boldsymbol{W}_1 + \boldsymbol{W}_2\delta\boldsymbol{W}_1\right)\right]\right\rangle.
\end{aligned} \tag{64}$$

The first noise term of the projection, $\phi$, is

$$\begin{aligned}
& \frac{-1}{L_xL_y}\left\langle\mathrm{tr}[(\boldsymbol{E}^T + \boldsymbol{W}_1^T\boldsymbol{W}_2^T)\delta\boldsymbol{W}_2\delta\boldsymbol{W}_1]\right\rangle \\
& \approx \frac{-2\eta^2}{L_xL_y}\left(\mathrm{tr}[\boldsymbol{W}_1^T\boldsymbol{W}_2^T\boldsymbol{E}]\mathrm{tr}[(\boldsymbol{E}^T + \boldsymbol{W}_1^T\boldsymbol{W}_2^T)\boldsymbol{E}] + \sigma_t^2\mathrm{tr}[(\boldsymbol{E}^T + \boldsymbol{W}_1^T\boldsymbol{W}_2^T)\boldsymbol{W}_2\boldsymbol{W}_1]\right) \\
& \approx 2\eta^2\left(L_xL_y\phi[\epsilon - \phi] + \sigma_t^2[\phi - \psi]\right),
\end{aligned} \tag{65}$$

where

$$\psi \equiv \frac{1}{L_xL_y}\mathrm{tr}[\boldsymbol{W}_1^T\boldsymbol{W}_2^T\boldsymbol{W}_2\boldsymbol{W}_1] = \frac{1}{L_xL_y}\|\boldsymbol{A}\|_F^2 - (\epsilon + 2\phi), \tag{66}$$

and the rest of the noise terms are the same as those for $\langle\delta\epsilon\rangle$. Thus,

$$\begin{aligned}
\langle\delta\phi\rangle \approx & \eta(\alpha + \beta)(\epsilon - \phi) + 2\eta^2\left(L_xL_y\phi[\epsilon - \phi] - \sigma_t^2[\psi - \phi]\right) \\
& - \eta^2\left(L_y[1 + \beta][\alpha_2 + \beta] + 4\alpha\beta\right)(L_x\epsilon + \sigma_t^2).
\end{aligned} \tag{67}$$

In summary, the learning dynamics of NP is described by following order parameter dynamics:

$$\langle\delta\alpha\rangle \approx 2\eta L_y\phi + \eta^2 L_hL_y(1 + \beta)(L_x\epsilon + \sigma_t^2), \tag{68a}$$

$$\langle\delta\beta\rangle \approx 2\eta L_x\phi + \eta^2 L_xL_y\alpha(1 + \beta)(L_x\epsilon + \sigma_t^2), \tag{68b}$$

$$\langle\delta\epsilon\rangle \approx -2\eta(\alpha + \beta)\epsilon - 4\eta^2\phi(L_xL_y\epsilon + \sigma_t^2) + \eta^2\left(L_y[1 + \beta][\alpha^2 + \beta] + 4\alpha\beta\right)(L_x\epsilon + \sigma_t^2), \tag{68c}$$

$$\begin{aligned}
\langle\delta\phi\rangle \approx & \eta(\alpha + \beta)(\epsilon - \phi) + 2\eta^2\left(L_xL_y\phi[\epsilon - \phi] - \sigma_t^2[\psi - \phi]\right) \\
& - \eta^2\left(L_y[1 + \beta][\alpha^2 + \beta] + 4\alpha\beta\right)(L_x\epsilon + \sigma_t^2).
\end{aligned} \tag{68d}$$

By tracking the dynamics of the order parameters, we can accurately predict the minimum training time required for reaching a target performance (dotted lines in Figs. S2A and S2B).

### B.3 Learning dynamics under Xavier-Glorot initialization

Under the Xavier-Glorot initialization [37], the initial weights are given by

$$W_{ij}^{(1)} \sim N\left(0, \frac{2}{L_x + L_h}\right),\ W_{ij}^{(2)} \sim N\left(0, \frac{2}{L_h + L_y}\right). \tag{69}$$

We also assume that the teacher weights follow the same random initialization:

$$A_{ij} \sim N\left(0, \frac{2}{L_x + L_y}\right). \tag{70}$$

Then, at the initial state, $\alpha$, $\beta$, and $\epsilon$ are $\mathcal{O}(1)$ with respect to $L_h$, while $\phi$ is $\mathcal{O}(L_h^{-1})$:

$$\alpha_{init} \approx \frac{2L_h}{L_x + L_h}, \; \beta_{init} \approx \frac{2L_h}{L_h + L_y},$$

$$\epsilon_{init} \approx \frac{4L_h}{(L_x + L_h)(L_h + L_y)} + \frac{2}{L_x + L_y}, \; \phi_{init} \approx \frac{-4L_h}{(L_x + L_h)(L_h + L_y)}. \tag{71}$$

Equations above follow from Eq. 44, by approximating the summation over weights with the expectation over Gaussian random variables. Notably, the initial output of the student model scales with $\mathcal{O}\left(L_h^{-1/2}\right)$, which is significantly smaller than the target output (unless $L_y \gg L_x$).

Let us reparameterize $\eta$ and $\alpha$ as

$$\eta = \eta_o L_h^{-\rho}, \; \alpha = a L_h^{\kappa}, \tag{72}$$

with $\rho > 0$ and $\kappa > 0$, to separate out the $L_h$ dependence of both variables. By substituting $\eta$ and $\alpha$ with the equations above, and ignoring terms that disappear at the large $L_h$ limit, we arrive at

$$\delta a \approx 2\eta_o L_h^{-(\rho+\kappa)} L_y \phi + \eta_o^2 L_h^{1-(2\rho+\kappa)} L_y (1+\beta)(L_x \epsilon + \sigma_t^2), \tag{73a}$$

$$\delta\beta \approx \eta_o L_h^{-\rho} L_x \phi + \eta_o^2 L_h^{-(2\rho-\kappa)} L_x L_y a(1+\beta)(L_x \epsilon + \sigma_t^2), \tag{73b}$$

$$\delta\epsilon \approx -2\eta_o L_h^{-(\rho-\kappa)} a\epsilon + \eta_o^2 L_h^{-2(\rho-\kappa)} L_y a^2(1+\beta)(L_x \epsilon + \sigma_t^2), \tag{73c}$$

$$\delta\phi \approx \eta_o L_h^{-(\rho-\kappa)} a(\epsilon - \phi) - \eta_o^2 L_h^{-2(\rho-\kappa)} L_y a^2(1+\beta)(L_x \epsilon + \sigma_t^2). \tag{73d}$$

Importantly, the first equation indicates that $1 - (2\rho + \kappa) \leq 0$ is necessary for preventing $a$ from exploding, while from the third equation, $\delta\epsilon < 0$ requires $\rho \geq \kappa$. To achieve the fastest convergence, we should use the largest learning rate. Hence, we should choose the smallest stable $\rho$. The smallest $\rho$ that satisfies the two conditions above is $\rho = \frac{1}{3}$, at which $\kappa = \frac{1}{3}$. Then, assuming $\phi \sim \mathcal{O}(1)$, we get $\delta\beta \sim \mathcal{O}(L_h^{-1/3})$. Thus, unless the number of weight updates satisfies $n_{th} > \mathcal{O}(L_h^{1/3})$, $\beta$ can be regarded as a constant. We can also ignore $\phi$, because it doesn't influence the dynamics of other variables in this limit. Therefore, the dynamics of the error is captured by the following two equations.

$$a_{n+1} = a_n + \eta_o^2 L_y(1+\beta_o)(L_x \epsilon_n + \sigma_t^2), \tag{74a}$$

$$\epsilon_{n+1} = \epsilon_n - 2\eta_o a_n \epsilon_n + \eta_o^2 L_y(1+\beta_o)a_n^2(L_x \epsilon_n + \sigma_t^2). \tag{74b}$$

Notably, the $L_h$ dependence disappears in the above equations, which indicates that the minimum learning time is independent of $L_h$.

In the limit of $\eta_o \to 0$, the update rules derived above can be written down as a set of differential equations. Moreover, as we show below, when $\sigma_t^2 = 0$, we can obtain an analytical expression of the minimum learning time because there is a conserved quantity in the differential equations. Hence, below we first consider $\eta_o \to 0$ and $\sigma_t^2 = 0$ limit, then study the effect of non-zero noise later. In this limit, the update rules become,

$$\dot{a} = \eta_o L_x L_y(1+\beta_o)\epsilon, \tag{75a}$$

$$\dot{\epsilon} = -(2a - \eta_o L_x L_y a^2[1+\beta_o])\epsilon. \tag{75b}$$

Denoting

$$c_o \equiv \eta_o L_x L_y(1+\beta_o), \tag{76}$$

these equations become

$$\dot{a} = c_o \epsilon, \tag{77a}$$

$$\dot{\epsilon} = -(2 - c_o a)a\epsilon. \tag{77b}$$

This dynamical system has the following conserved quantity:

$$\frac{d}{dt}\left(\epsilon + \frac{1}{c_o}a^2 - \frac{1}{3}a^3\right) = 0. \tag{78}$$

This means that $a$ follows

$$\dot{a} = c_o\left(\epsilon_{tot} - \frac{1}{c_o}a^2 + \frac{1}{3}a^3\right), \tag{79}$$

where $\epsilon_{tot}$ is a constant that only depends on the initial values of $\epsilon$ and $a$. Therefore, the minimum learning time $T_{np}$, required for reaching target error $\epsilon_{tg}$ from initial error $\epsilon_{init}$, is given as

$$T_{np} = \frac{1}{\eta_o}\int_{a_{init}}^{a_{tg}}\frac{da}{c_o\epsilon_{tot} + (c_o/3)a^3 - a^2}, \tag{80}$$

where $a_{init}$ and $a_{tg}$ are the normalized input weight norm at $\epsilon = \epsilon_{init}$ and $\epsilon = \epsilon_{tg}$, respectively:

$$\epsilon_{tot} = \epsilon_{init} - \frac{(a_{init})^3}{3} + \frac{(a_{init})^2}{c_o} = \epsilon_{tg} - \frac{(a_{tg})^3}{3} + \frac{(a_{tg})^2}{c_o}. \tag{81}$$

In Eq. 13 in the main text, we used $\frac{1}{\eta_o} = L_x L_y \frac{1+\beta_o}{c_o}$, which follows from Eq. 76. Moreover, for achieving a convergence without a resurgence of the error, the weight norm $a$ has to satisfy $a_{tg} < 2/c_o$ at $\epsilon = \epsilon_{tg}$. This means that the coefficient $c_o$ should satisfy

$$\epsilon_{tot} = \epsilon_{init} + \frac{a_{init}^2}{c_o} - \frac{a_{init}^3}{3} < \epsilon_{tg} + \frac{1}{c_o}\left(\frac{2}{c_o}\right)^2 - \frac{1}{3}\left(\frac{2}{c_o}\right)^3 = \epsilon_{tg} + \frac{4}{3c_o^3},$$

$$\Leftrightarrow \left([\epsilon_{init} - \epsilon_{tg}] - \frac{a_{init}^3}{3}\right)c_o^3 + a_{init}^2 c_o^2 - \frac{4}{3} < 0. \tag{82}$$

Denoting the positive root of this equation as $c_o^*$, the critical learning rate $\eta^*$ is given as

$$\eta^* = \frac{c_o^* L_h^{-1/3}}{L_x L_y(1 + \beta_o)}. \tag{83}$$

Numerically estimated optimal learning rates match well with this prediction (Fig. S2B). Note that the optimal learning rate $\hat{\eta}$ that minimizes the training time becomes roughly the same as the critical learning rate $\eta^*$ in this setting because the norm $a$ expands rapidly during learning.

In the presence of noise (ie $\sigma_t^2 > 0$), on the contrary, the learning dynamics approximately follows

$$\dot{a} = c_o(\epsilon + \sigma_t^2/L_x), \tag{84a}$$

$$\dot{\epsilon} = -2a\epsilon + c_o a^2(\epsilon + \sigma_t^2/L_x). \tag{84b}$$

Thus, the nullcline of the error $\epsilon$ is given as

$$\epsilon = \frac{c_o a \sigma_t^2}{L_x(2 - c_o a)}. \tag{85}$$

## B.4 Mean-field dynamics under SGD

Under SGD, the weight dynamics follows

$$\delta W_1 = -\eta W_2^T\left(Exx^T + \sigma_t zx^T\right), \tag{86}$$

$$\delta W_2 = -\eta\left(Exx^T + \sigma_t zx^T\right)W_1^T. \tag{87}$$

Hence, using the approximation introduced in Eqs. 52 and 57, we get the mean-field description as below. First, taking the expectation over $x$ and $z$, the dynamics of $\alpha$ is given as

$$\langle\delta\alpha\rangle = \frac{-2\eta}{L_x}\mathrm{tr}[W_1^T W_2^T E] + \frac{\eta^2}{L_x}\left\langle\mathrm{tr}[x(x^T E^T + \sigma_t z^T)W_2 W_2^T(Ex + \sigma_t z)x^T]\right\rangle$$

$$= \frac{-2\eta}{L_x}\mathrm{tr}[W_1^T W_2^T E] + \eta^2\mathrm{tr}[W_2 W_2^T(EE^T + \sigma_t^2 I)]$$

$$\approx 2\eta L_y\phi + \eta^2 L_y\beta(L_x\epsilon + \sigma_t^2). \tag{88}$$

Similarly, the dynamics of $\beta$ follows

$$\langle\delta\beta\rangle = \frac{-2\eta}{L_y}\mathrm{tr}[\boldsymbol{W}_2^T\boldsymbol{E}\boldsymbol{W}_1^T] + \frac{\eta^2}{L_y}\left\langle\mathrm{tr}[\boldsymbol{W}_1\boldsymbol{x}(\boldsymbol{x}^T\boldsymbol{E}^T + \sigma_t\boldsymbol{z}^T)(\boldsymbol{E}\boldsymbol{x} + \sigma_t\boldsymbol{z})\boldsymbol{x}^T\boldsymbol{W}_1^T]\right\rangle$$
$$\approx 2\eta L_x\phi + \eta^2 L_x\alpha(L_x\epsilon + \sigma_t^2). \tag{89}$$

As before, up to the second-order term, the error variable $\epsilon$ follows

$$\delta\epsilon = \frac{2}{L_xL_y}\mathrm{tr}[\boldsymbol{E}^T(\delta\boldsymbol{W}_2\boldsymbol{W}_1 + \boldsymbol{W}_2\delta\boldsymbol{W}_1)] + \frac{2}{L_xL_y}\mathrm{tr}[\boldsymbol{E}^T\delta\boldsymbol{W}_2\delta\boldsymbol{W}_1]$$
$$+ \frac{1}{L_xL_y}\mathrm{tr}[(\delta\boldsymbol{W}_2\boldsymbol{W}_1 + \boldsymbol{W}_2\delta\boldsymbol{W}_1)^T(\delta\boldsymbol{W}_2\boldsymbol{W}_1 + \boldsymbol{W}_2\delta\boldsymbol{W}_1)], \tag{90}$$

where each term is estimated as

$$\frac{2}{L_xL_y}\left\langle\mathrm{tr}[\boldsymbol{E}^T(\delta\boldsymbol{W}_2\boldsymbol{W}_1 + \boldsymbol{W}_2\delta\boldsymbol{W}_1)]\right\rangle \approx -2\eta(\alpha + \beta)\epsilon, \tag{91a}$$

$$\frac{2}{L_xL_y}\left\langle\mathrm{tr}[\boldsymbol{E}^T\delta\boldsymbol{W}_2\delta\boldsymbol{W}_1]\right\rangle = \frac{2\eta^2}{L_xL_y}\left\langle\mathrm{tr}[\boldsymbol{E}^T(\boldsymbol{E}\boldsymbol{x} + \sigma_t\boldsymbol{z})\boldsymbol{x}^T\boldsymbol{W}_1^T\boldsymbol{W}_2^T(\boldsymbol{E}\boldsymbol{x} + \sigma_t\boldsymbol{z})\boldsymbol{x}^T]\right\rangle$$
$$\approx -2\eta^2(L_xL_y\epsilon + \sigma_t^2)\phi, \tag{91b}$$

$$\frac{1}{L_xL_y}\left\langle\mathrm{tr}[\boldsymbol{W}_1^T\delta\boldsymbol{W}_2^T\delta\boldsymbol{W}_2\boldsymbol{W}_1]\right\rangle = \frac{\eta^2}{L_xL_y}\left\langle\mathrm{tr}[\boldsymbol{W}_1^T\boldsymbol{W}_1\boldsymbol{x}(\boldsymbol{x}^T\boldsymbol{E}^T\boldsymbol{E}\boldsymbol{x} + \sigma_t^2\boldsymbol{z}^T\boldsymbol{z})\boldsymbol{x}^T\boldsymbol{W}_1^T\boldsymbol{W}_1]\right\rangle$$
$$\approx \eta^2(L_x\epsilon + \sigma_t^2)\alpha_2, \tag{91c}$$

$$\frac{1}{L_xL_y}\left\langle\mathrm{tr}[\delta\boldsymbol{W}_1^T\boldsymbol{W}_2^T\boldsymbol{W}_2\delta\boldsymbol{W}_1]\right\rangle = \frac{\eta^2}{L_xL_y}\left\langle\mathrm{tr}[\boldsymbol{W}_2^T\boldsymbol{W}_2\boldsymbol{W}_2^T(\boldsymbol{E}\boldsymbol{x}\boldsymbol{x}^T\boldsymbol{x}\boldsymbol{x}^T\boldsymbol{E} + \sigma_t^2\boldsymbol{z}\boldsymbol{x}^T\boldsymbol{x}\boldsymbol{z}^T)\boldsymbol{W}_2]\right\rangle$$
$$\approx \eta^2(L_x\epsilon + \sigma_t^2)\beta_2, \tag{91d}$$

$$\frac{2}{L_xL_y}\left\langle\mathrm{tr}[\delta\boldsymbol{W}_1^T\boldsymbol{W}_2^T\delta\boldsymbol{W}_2\boldsymbol{W}_1]\right\rangle = \frac{2}{L_xL_y}\left\langle\mathrm{tr}[\boldsymbol{x}^T(\boldsymbol{x}^T\boldsymbol{E}^T + \sigma_t\boldsymbol{z}^T)\boldsymbol{W}_2\boldsymbol{W}_2^T(\boldsymbol{E}\boldsymbol{x} + \sigma_t\boldsymbol{z})\boldsymbol{x}^T\boldsymbol{W}_1^T\boldsymbol{W}_1]\right\rangle$$
$$\approx 2\eta^2\alpha\beta(L_x\epsilon + \sigma_t^2). \tag{91e}$$

Here, we defined $\beta_2$ by $\beta_2 \equiv \frac{1}{L_y}\mathrm{tr}[\boldsymbol{W}_2\boldsymbol{W}_2^T\boldsymbol{W}_2\boldsymbol{W}_2^T]$. As in the case of $\alpha_2$, we approximate $\beta_2$ with $\beta^2$ below (see Eq. 94). Summing up,

$$\langle\delta\epsilon\rangle \approx -2\eta(\alpha + \beta)\epsilon - 2\eta^2(L_xL_y\epsilon + \sigma_t^2)\phi + \eta^2(L_x\epsilon + \sigma_t^2)(\alpha_2 + 2\alpha\beta + \beta_2). \tag{92}$$

Similarly, taking the expectation over $\boldsymbol{x}$ and $\boldsymbol{z}$, the dynamics of $\phi$ follows

$$\langle\delta\phi\rangle \approx \eta(\alpha + \beta)(\epsilon - \phi) + \eta^2(L_xL_y\phi[\epsilon - \phi] - \sigma_t^2[\psi - \phi]) - \eta^2(L_x\epsilon + \sigma_t^2)(\alpha_2 + 2\alpha\beta + \beta_2), \tag{93}$$

where $\psi$ is defined in Eq. 66. Noticing that, so far $\alpha_2$ and $\beta_2$ only appeared on the noise term of $\langle\delta\epsilon\rangle$ and $\langle\delta\phi\rangle$ as $\alpha_2 + 2\alpha\beta + \beta_2$, by introducing an additional approximation:

$$\alpha_2 + 2\alpha\beta + \beta_2 \approx (\alpha + \beta)^2, \tag{94}$$

we get a closed-form dynamics of $\alpha$,$\beta$, $\epsilon$, and $\phi$, from Eqs. 88, 89, 92, 93, and 66.

Because $\langle\delta\alpha\rangle$ doesn't depend on $L_h$ (Eq. 88), we regard $\alpha$ as a constant, $\alpha \approx \alpha_{init}$, in addition to $\beta \approx \beta_{init}$. Moreover, because $\phi_{init} \sim \mathcal{O}(1/L_h)$, the second term of Eq. 92 is negligible. Therefore, in the $\sigma_t^2 = 0$ limit, the error dynamics follows

$$\langle\delta\epsilon\rangle \approx -2\eta(\alpha_{init} + \beta_{init})\epsilon + \eta^2(\alpha_{init} + \beta_{init})^2 L_x\epsilon. \tag{95}$$

Thus, the optimal learning rate and the minimum training time are given as

$$\hat{\eta} = \frac{1}{L_x(\alpha_{init} + \beta_{init})}, \quad T_{sgd}[\epsilon_o \to \epsilon_{tg}] = L_x\log(\epsilon_o/\epsilon_{tg}). \tag{96}$$

## B.5 Random update approximation

Let us initialize $\boldsymbol{W}_1$ and $\boldsymbol{W}_2$ as random matrices, where the elements of $\boldsymbol{W}_1$ and $\boldsymbol{W}_2$ are sampled independently from zero-mean Gaussian distribution with the variance $\sigma_{(1)}^2$ and $\sigma_{(2)}^2$, respectively. Then, the error on the approximation Eq. 52 is estimated as

$$\left\langle \left( \mathrm{tr}[\boldsymbol{W}_2\boldsymbol{W}_2^T\boldsymbol{E}\boldsymbol{E}^T] - \frac{1}{L_y}\mathrm{tr}[\boldsymbol{W}_2\boldsymbol{W}_2^T]\mathrm{tr}[\boldsymbol{E}\boldsymbol{E}^T] \right)^2 \right\rangle_{\boldsymbol{W}_1,\boldsymbol{W}_2}$$

$$\leq 2\left\langle \left( \mathrm{tr}[\boldsymbol{W}_2\boldsymbol{W}_2^T\boldsymbol{A}\boldsymbol{A}^T] - \frac{1}{L_y}\mathrm{tr}[\boldsymbol{W}_2\boldsymbol{W}_2^T]\mathrm{tr}[\boldsymbol{A}\boldsymbol{A}^T] \right)^2 \right\rangle$$

$$+ 2\left\langle \left( \mathrm{tr}[\boldsymbol{W}_2\boldsymbol{W}_2^T\boldsymbol{W}_2\boldsymbol{W}_1\boldsymbol{W}_1^T\boldsymbol{W}_2^T] - \frac{1}{L_y}\mathrm{tr}[\boldsymbol{W}_2\boldsymbol{W}_2^T]\mathrm{tr}[\boldsymbol{W}_2\boldsymbol{W}_1\boldsymbol{W}_1^T\boldsymbol{W}_2^T] \right)^2 \right\rangle$$

$$+ 4\left\langle \left( \mathrm{tr}[\boldsymbol{W}_2\boldsymbol{W}_2^T\boldsymbol{W}_2\boldsymbol{W}_1\boldsymbol{A}^T] - \frac{1}{L_y}\mathrm{tr}[\boldsymbol{W}_2\boldsymbol{W}_2^T]\mathrm{tr}[\boldsymbol{W}_2\boldsymbol{W}_1\boldsymbol{A}^T] \right)^2 \right\rangle. \tag{97}$$

The first term of the equation above is further decomposed into

$$\left\langle \left( \mathrm{tr}[\boldsymbol{W}_2\boldsymbol{W}_2^T\boldsymbol{A}\boldsymbol{A}^T] - \frac{1}{L_y}\mathrm{tr}[\boldsymbol{W}_2\boldsymbol{W}_2^T]\mathrm{tr}[\boldsymbol{A}\boldsymbol{A}^T] \right)^2 \right\rangle$$

$$\leq 2\left\langle \left( \mathrm{tr}[\boldsymbol{W}_2\boldsymbol{W}_2^T\boldsymbol{A}\boldsymbol{A}^T] - \sum_i^{L_y}[\boldsymbol{W}_2\boldsymbol{W}_2^T]_{ii}[\boldsymbol{A}\boldsymbol{A}^T]_{ii} \right)^2 \right\rangle$$

$$+ 2\left\langle \left( \sum_i^{L_y}[\boldsymbol{W}_2\boldsymbol{W}_2^T]_{ii}[\boldsymbol{A}\boldsymbol{A}^T]_{ii} - \frac{1}{L_y}\mathrm{tr}[\boldsymbol{W}_2\boldsymbol{W}_2^T]\mathrm{tr}[\boldsymbol{A}\boldsymbol{A}^T] \right)^2 \right\rangle, \tag{98}$$

and the two terms are evaluated as

$$\left\langle \left( \mathrm{tr}[\boldsymbol{W}_2\boldsymbol{W}_2^T\boldsymbol{A}\boldsymbol{A}^T] - \sum_i^{L_y}[\boldsymbol{W}_2\boldsymbol{W}_2^T]_{ii}[\boldsymbol{A}\boldsymbol{A}^T]_{ii} \right)^2 \right\rangle$$

$$= \left\langle \left( \sum_i^{L_y}\sum_j^{L_h}\sum_{k\neq i}^{L_y}\sum_l^{L_x} A_{il}A_{kl}w_{ij}^{(2)}w_{kj}^{(2)} \right)^2 \right\rangle = \mathcal{O}\left( L_h\sigma_{(2)}^4 \right),$$

and

$$\left\langle \left( \sum_i^{L_y}[\boldsymbol{W}_2\boldsymbol{W}_2^T]_{ii}[\boldsymbol{A}\boldsymbol{A}^T]_{ii} - \frac{1}{L_y}\mathrm{tr}[\boldsymbol{W}_2\boldsymbol{W}_2^T]\mathrm{tr}[\boldsymbol{A}\boldsymbol{A}^T] \right)^2 \right\rangle$$

$$= \left\langle \left( \sum_i^{L_y}\sum_j^{L_h}\left( (w_{ij}^{(2)})^2 - \sigma_{(2)}^2 \right)\left( [\boldsymbol{A}\boldsymbol{A}^T]_{ii} - \frac{1}{L_y}\mathrm{tr}[\boldsymbol{A}\boldsymbol{A}^T] \right) \right)^2 \right\rangle$$

$$= \mathcal{O}\left( L_h\sigma_{(2)}^4 \right).$$

Applying the same decomposition, the second term of Eq. 97 satisfies

$$\left\langle \left( \mathrm{tr}[\boldsymbol{W}_2\boldsymbol{W}_2^T\boldsymbol{W}_2\boldsymbol{W}_1\boldsymbol{W}_1^T\boldsymbol{W}_2^T] - \frac{1}{L_y}\mathrm{tr}[\boldsymbol{W}_2\boldsymbol{W}_2^T]\mathrm{tr}[\boldsymbol{W}_2\boldsymbol{W}_1\boldsymbol{W}_1^T\boldsymbol{W}_2^T] \right)^2 \right\rangle = \mathcal{O}\left( L_h^3\sigma_{(1)}^4\sigma_{(2)}^8 \right).$$

Finally, the cross-term of Eq. 97 follows

$$\left\langle \left( \operatorname{tr}[\boldsymbol{W}_2 \boldsymbol{W}_2^T \boldsymbol{W}_2 \boldsymbol{W}_1 \boldsymbol{A}^T] - \frac{1}{L_y} \operatorname{tr}[\boldsymbol{W}_2 \boldsymbol{W}_2^T] \operatorname{tr}[\boldsymbol{W}_2 \boldsymbol{W}_1 \boldsymbol{A}^T] \right)^2 \right\rangle$$

$$\leq 2 \left\langle (\operatorname{tr}[\boldsymbol{W}_2 \boldsymbol{W}_2^T \boldsymbol{W}_2 \boldsymbol{W}_1 \boldsymbol{A}^T])^2 \right\rangle + \frac{2}{L_y^2} \left\langle (\operatorname{tr}[\boldsymbol{W}_2 \boldsymbol{W}_2^T] \operatorname{tr}[\boldsymbol{W}_2 \boldsymbol{W}_1 \boldsymbol{A}^T])^2 \right\rangle .$$

$$= \mathcal{O} \left( L_h^3 \sigma_{(1)}^2 \sigma_{(2)}^6 \right)$$

Therefore, we have

$$\operatorname{tr}[\boldsymbol{W}_2 \boldsymbol{W}_2^T \boldsymbol{E} \boldsymbol{E}^T] = \frac{1}{L_y} \operatorname{tr}[\boldsymbol{W}_2 \boldsymbol{W}_2^T] \operatorname{tr}[\boldsymbol{E} \boldsymbol{E}^T] + \mathcal{O} \left( \max \left\{ L_h^{1/2} \sigma_{(2)}^2, L_h^{3/2} \sigma_{(1)} \sigma_{(2)}^3 \right\} \right) . \tag{99}$$

Applying the same calculation for Eq. 57, we get

$$\operatorname{tr}[\boldsymbol{W}_1^T \boldsymbol{W}_1 \boldsymbol{E}^T \boldsymbol{E}] = \frac{1}{L_x} \operatorname{tr}[\boldsymbol{W}_1^T \boldsymbol{W}_1] \operatorname{tr}[\boldsymbol{E}^T \boldsymbol{E}] + \mathcal{O} \left( \max \left\{ L_h^{1/2} \sigma_{(1)}^2, L_h^{3/2} \sigma_{(1)}^3 \sigma_{(2)} \right\} \right) . \tag{100}$$

Therefore, under the Xavier-glorot initialization, where $\sigma_{(1)}^2 = \mathcal{O} \left( L_h^{-1} \right)$ and $\sigma_{(1)}^2 = \mathcal{O} \left( L_h^{-1} \right)$, these two approximations hold. More precisely, if $\sigma_{(2)}^2 = \mathcal{O} \left( L_h^{-1} \right)$, then $\sigma_{(1)}^2 = o \left( L_h^{-2/3} \right)$ is sufficient for the approximation.

Suppose we add up random matrices $\delta \boldsymbol{W}_1$ and $\delta \boldsymbol{W}_2$ onto $\boldsymbol{W}_1$ and $\boldsymbol{W}_2$, where the elements of $\delta \boldsymbol{W}_1$ and $\delta \boldsymbol{W}_2$ are sampled from a zero-mean Gaussian distribution. Then, as long as the variances of $\boldsymbol{W}_1$ and $\boldsymbol{W}_2$ satisfy the conditions above, these approximations is still valid. Although the weight updates $\delta \boldsymbol{W}_1$ and $\delta \boldsymbol{W}_2$ in NP are dominated by Gaussian noise from the perturbations, they are not random. Thus, these approximations cease to hold eventually.

## B.6  NP and SGD in linear regression

Here, we briefly sketch the analysis of SGD and NP in linear regression setting, studied in [12]. Let us consider a linear regression problem where the teacher network is given as $\boldsymbol{y}^* = \boldsymbol{A}\boldsymbol{x}$, while the student network is $\boldsymbol{y} = \boldsymbol{W}\boldsymbol{x}$, and input $\boldsymbol{x}$ is sampled independently from Gaussian distribution $x_i \sim N(0, 1)$. Given L-2 loss $\epsilon = \frac{1}{2} \left\langle \|\boldsymbol{y} - \boldsymbol{y}^*\|^2 \right\rangle = \frac{1}{2} \|\boldsymbol{W} - \boldsymbol{A}\|_F^2$, SGD and NP updates are given by

$$\delta W^{sgd} = -\eta(\boldsymbol{W} - \boldsymbol{A})\boldsymbol{x}\boldsymbol{x}^T, \tag{101a}$$

$$\delta W^{np} = -\eta \boldsymbol{\xi} \boldsymbol{\xi}^T (\boldsymbol{W} - \boldsymbol{A})\boldsymbol{x}\boldsymbol{x}^T. \tag{101b}$$

Let us assume that the size of input and output layers, $L_x$ and $L_y$, satisfy $L_x, L_y \gg 1$. Then, after $t$ steps of training, the loss $\epsilon^{sgd}$ and $\epsilon^{np}$ follow (see [12] for the details)

$$\epsilon_t^{sgd} = (1 - 2\eta + L_x \eta^2)^t \epsilon_o^{sgd}, \tag{102a}$$

$$\epsilon_t^{np} = (1 - 2\eta + L_x L_y \eta^2)^t \epsilon_o^{np}. \tag{102b}$$

Therefore, the training time from the initial error $\epsilon_o$ to a target error $\epsilon_{tg}$ is given as

$$T_{sgd} [\epsilon_o \rightarrow \epsilon_{tg}] = \frac{\log (\epsilon_{tg}/\epsilon_o)}{\log (1 - 2\eta + L_x \eta^2)}, \quad T_{np} [\epsilon_o \rightarrow \epsilon_{tg}] = \frac{\log (\epsilon_{tg}/\epsilon_o)}{\log (1 - 2\eta + L_x L_y \eta^2)}. \tag{103}$$

At the optimal learning rates that minimize the training time, $\hat{\eta}_{sgd} = \frac{1}{L_x}$ and $\hat{\eta}_{np} = \frac{1}{L_x L_y}$, the training times follow:

$$T_{sgd} [\epsilon_o \rightarrow \epsilon_{tg}] = L_x \log (\epsilon_o/\epsilon_{tg}), \quad T_{np} [\epsilon_o \rightarrow \epsilon_{tg}] = L_x L_y \log (\epsilon_{tg}/\epsilon_o). \tag{104}$$

Please note that, the critical learning rates, the largest stable rates, are twice larger than the optimal rates in linear regression: $\eta_{sgd}^* = \frac{2}{L_x}$ and $\eta_{np}^* = \frac{2}{L_x L_y}$.

# C   Model setting details

## C.1   Model implementation

### Implementation of NP

We implemented the node perturbation algorithm using Eq. 3. In nonlinear networks, we also trained bias parameters by

$$\delta \boldsymbol{b}_k^{np} = -\frac{\eta}{\sigma} \left( \ell(\widetilde{\boldsymbol{x}}_K, \boldsymbol{x}_0) - \ell(\boldsymbol{x}_K, \boldsymbol{x}_0) \right) \boldsymbol{\xi}_k, \tag{105}$$

where $\boldsymbol{b}_k$ represent the bias of the $k$-th layer. In both regression and classification tasks, we defined the loss by the squared error: $\ell(\boldsymbol{x}_K, \boldsymbol{x}_K^*) = \frac{1}{2} \|\boldsymbol{x}_K - \boldsymbol{x}_K^*\|^2$, where $\boldsymbol{x}_K^*$ is the supervised output. We set the perturbation amplitude $\sigma$ to $\sigma = 10^{-6}$ in deep linear models, and $\sigma = 10^{-4}$ in deep nonlinear models.

### Student-teacher model

In the simulations, we generated the teacher network randomly with

$$A_{ij} \sim N\left(0, \frac{2}{L_x + L_y}\right). \tag{106}$$

We initialized the weights $\boldsymbol{W}_1, \boldsymbol{W}_2$ of the student network with the Xavier-Glorot initialization [37]:

$$W_{ij}^{(1)} \sim N\left(0, \frac{2}{L_x + L_h}\right), \ W_{ij}^{(2)} \sim N\left(0, \frac{2}{L_y + L_h}\right). \tag{107}$$

### Weight normalization

In both linear and nonlinear networks, we implemented weight normalization by

$$\boldsymbol{w}_i^k \rightarrow \frac{\|\boldsymbol{w}_i^k\|}{\|\boldsymbol{w}_i^k + \delta \boldsymbol{w}_i^k\|}(\boldsymbol{w}_i^k + \delta \boldsymbol{w}_i^k), \tag{108}$$

where $\boldsymbol{w}_i^k$ is the input weight vector of $i$-th node in the $k$-th layer, and $\delta \boldsymbol{w}_i^k$ is the weight update calculated by either NP or SGD. Note that, this normalization biases the update from the original update $\delta \boldsymbol{w}_i^k$. In deep nonlinear networks, we applied the normalization only to the weight parameters, not to the bias parameters.

It should be also noted that the neuron-wise weight normalization is rewritten as

$$\boldsymbol{w}_i^k \rightarrow \boldsymbol{w}_i^k - \left(1 - \frac{\|\boldsymbol{w}_i^k\|}{\|\boldsymbol{w}_i^k + \delta \boldsymbol{w}_i^k\|}\right) \boldsymbol{w}_i^k + \frac{\|\boldsymbol{w}_i^k\|}{\|\boldsymbol{w}_i^k + \delta \boldsymbol{w}_i^k\|} \delta \boldsymbol{w}_i^k. \tag{109}$$

Therefore, the weight normalization we employed can be considered as adaptive weight decay, which is largely consistent with experimentally-observed synaptic scaling mechanisms [46].

### Linear dimensionality

We calculated the linear dimensionality by the participation ratio: $\mathrm{tr}[\Sigma_h]^2 / \mathrm{tr}[\Sigma_h^2]$, where $\Sigma_h$ is the covariance of the hidden layer activity over the training dataset. Analyzing the eigenspectrum of the covariance matrix, we found that the contribution of the principal eigenvalue is larger under NP than SGD. This is consistent with excessive weight expansion under NP, because the principal eigen-component of ReLU units activity typically captures overall non-negativity of the activity, which is modulated by the weight norm.

### Code availability

Simulation codes are made available at `https://github.com/nhiratani/node_perturbation`. All simulations were performed on standard laboratory CPUs and GPUs.

## C.2 Figure details

### Figure 1

Neural architecture: 100-10000-10.
Activation: Linear (all layers).
Task: Student-teacher with zero noise (see Appendix B.1).

In panel **A**, the solid lines are the average trajectory over 10 independent simulations. In panels **B** and **C**, we depicted learning trajectories under ten different random seeds for each learning rate.

### Figure 2

Neural architecture: 100-$L_h$-10 (**A**), 100-10000-$L_y$ (**B**).
Activation: Linear (all layers).
Task: Student-teacher with zero noise.

In both panels **A** and **B**, the target error was set to $\epsilon_{tg} = 10^{-4}\epsilon_o$. In the simulation, we estimated the minimum training time by optimizing the learning rate $\eta$ between $10^{-2}\eta^* \leq \eta \leq 10\eta^*$, where $\eta^*$ is the analytically estimated critical learning rate (Eq. 83). For each learning rate, we estimated the minimum training time by taking the average over ten random seeds.

### Figure 3

Neural architecture: 100-10000-10 (**A, B, C**), 100-$L_h$-10 (**D**).
Activation: Linear (all layers).
Task: Student-teacher with noise $\sigma_t^2 = 0.1$.

In panel **A**, the purple line stops in the middle because we terminated the simulation after $10^6$ iterations. Both in panels **A** and **B**, $\eta^*$ is the critical learning rate in the absence of noise ($\sigma_t^2 = 0$). In panel **D**, the learning rate was set to the critical learning rate under each hidden layer size. In **D**, weight normalization was introduced to every neurons at every update. The learning trajectories in panels **A** and **D** are the average trajectories over 10 random seeds.

### Figure 4

Neural architecture: 21-$L_h$-7 (**A, E, F**), 21-800-7 (**B, C, D**).
Activation: ReLU (the hidden layer), Linear (the last layer),
Task: SARCOS.

In all simulations, we set the mini-batch size to 100, and initialized the weights by the Xavier-Glorot initialization [37]. In panel **A**, we trained the network with SGD until the error reaches $\epsilon = 50.0$ (solid lines) and $\epsilon = 5.0$ (dashed lines), then calculated the cosine similarity between SGD and NP updates. In panel **E**, We first estimated the average learning trajectory by binning the data over 10 consecutive epochs, and taking average over 5 simulation with different random seeds. We then defined the required number of epochs by the point at which the average error goes down below the threshold for the first time. We calculated this required number of epochs under various different learning rates (see Fig. S3B), then defined the minimum training time as the minimum among them. For SGD, we did not bin the data over epochs. In panel **F**, we used $\eta = 3.6 \times 10^{-5}, 2.4 \times 10^{-5}, 1.8 \times 10^{-5}$ for $L_h = 400, 800, 1600$, respectively, to make sure that the error diverges around the same time in the absence of the weight normalization. Here (in panel **F**), we initialized the input weight $W_1$, ten times larger than the standard Xavier-Glorot initialization. We observed qualitatively the same dynamics under the standard Xavier-Glorot initialization, but the convergence under the standard initialization was slightly worse.

### Figure 5

Neural architecture: 784-800-10 (**A**), 784-$L_h$-10 (**B**), 784-300-10 to 784-300-300-300-300-10 (**C**).
Activation: ReLU (the hidden layers), Softmax (the last layer).
Task: MNIST.

In all simulations, we set the mini-batch size to 1000, and initialized the weights by the Xavier-Glorot initialization. In panels **B** and **C**, the minimum training time was estimated in the same manner as in Fig. 4**E**.

### Figure S1

Neural architecture: 100-10
Activation: Linear
Task: Linear regression with $\sigma_t = 0$. The teacher model was generated randomly via Eq. 106.

### Figure S2

Neural architecture: 100-$L_h$-10.
Activation: Linear (all layers).
Task: Student-teacher with zero-noise.

### Figure S3

Neural architecture: 21-800-7 (**A,C,D**), 21-$L_h$-7 (**B**).
Activation: ReLU (the hidden layer), Linear (the last layer).
Task: SARCOS.

In all simulations, we set the mini-batch size to 100, and initialized the weights by the Xavier-Glorot initialization. In panel **C** and **D**, we set the learning rate of both SGD and NP to $\eta = 10^{-6}$.

### Figure S4

Neural architecture: 784-$L_h$-10 (**A**), 784-800-10 (**B-E**), 784-300-300-300-10 (**F**).
Activation: ReLU (the hidden layer), Softmax (the last layer).
Task: MNIST.

As in the main text, in all simulations, we set the mini-batch size to 1000, and initialized the weights by the Xavier-Glorot initialization. In panel **C**, we evaluated the error by the mean-squared error on the test data, and estimated the points at which the error becomes minimum (black points) and the error returns to the initial level ($\epsilon = 0.15$; gray points).

### Figure S5

Neural architecture: Three convolutional layer with 32 channels each and one fully connected layer.
Activation: ReLU (the hidden layers), Sigmoid (the last layer)
Task: CIFAR-10.