# OpenReview forum: "On the Stability and Scalability of Node Perturbation Learning"
_NeurIPS.cc/2022/Conference — NeurIPS 2022 Accept_

### Official Review · Reviewer_bpjP · 2022-06-26

**Rating:** 6
**Confidence:** 4
**Soundness:** 4 excellent
**Presentation:** 3 good
**Contribution:** 3 good

**Summary:**

The authors study a stochastic gradient-free learning rule for deep feed forward neural networks, node perturbation. The learning rule in brief is to perturb the activity of each neuron by a Gaussian and then update the corresponding weights in proportion to the sampled difference in the loss function.

The authors develop a mean field theory for a two-layered linear network whose target function is given by a linear transformation with additive Gaussian noise. This allows them to solve for a critical learning rate in which the learning process is as fast as possible. They find that even with the critical learning rate, node perturbation is slower than stochastic gradient descent. In particular, it is asymptotically slower by approximately a factor of the output size. In addition, their theoretical results reveal an instability in the error dynamics once a certain threshold is crossed, and the authors introduce weight regularization to remedy this.

Finally, the authors numerically test the performance of the regularized node perturbation algorithm on the supervised learning tasks MNIST and SARCOS, and show that the algorithm is indeed capable of learning these tasks.

**Questions:**

My most central and concrete question is whether NP is slower or faster than the reinforcement algorithm. I got mixed messages from the figures and from the discussion section.

In the abstract of Fiete et al, 2007, the authors state that their version of node perturbation "learns in a reasonable amount of time" and "Varying the number of neurons in HVC and RA has little effect on learning time." Do these 2007 results agree or disagree with the present results and why?

**Limitations:**

Societal impact is not very relevant; the results of the paper are far too theoretical for negative societal impact to be speculated upon. Performance limitations were adequately reflected upon.

**Strengths And Weaknesses:**

The authors provide original theoretical and numerical results about a particular learning algorithm which are clear and straightforward to understand. From a machine learning perspective, the main drawback of the paper is that they are mostly showing weaknesses of an existing algorithm instead of introducing something new and performant. That said, the weight regularization does successfully stabilize the learning process and it is somewhat interesting that the algorithm is able to learn MNIST and SARCOS with the regularization in place.

One problem I have with the SARCOS figures is that there is no benchmark shown; everyone knows that an accuracy of 98% on MNIST is acceptable, but since I'm not intimately familiar with the SARCOS task I would like to see either a SoTA MSE taken from the literature, or at least the MSE of the SGD-trained models alongside the NP-trained models for comparison. It is explained in the text that the target error was 5.0, but it is still not clear to me whether or not this is a good value in the grand scheme of algorithms which solve this task. Another way to make the error values easier to interpret would be to report $R^2$ instead of MSE, but either way some kind of benchmark should be shown.

The central justification for the lacking performance of the algorithm is its biological plausibility. This makes sense, but I did not get a very clear message from the authors as to whether or not they really believe this algorithm is used somewhere in the brain. I agree with the author's statement "an algorithm is biologically implausible if it takes an inordinately long time to reach good performance." However, I did not understand the citation of Figures 4E and 5B when they said "its [NP's] performance deficit is smaller when compared to
a reinforcement learning rule using the error backpropagation." From what I understood, these figures show that it takes longer to train NP than reinforcement learning or SGD, especially as depth increases. If I understood correctly, it seems this would be in contradiction to what is said in the discussion, and then it is difficult to rationalize the use of NP in the brain. There should be some clarification either in the figures or in the discussion.

I also think its worth noting that inverse problems (including SARCOS) are often solved innately by the brain. When a calf (or any quadruped) is born, they don't spend any number of epochs learning what torques are required to move their hooves from one place to the other, walking occurs immediately, suggesting that the capability was genetically encoded into the architecture, not learned. The authors could, of course, defend the claim that NP is used for learning more complex motor tasks, such as those performed by humans, but then it would also have to be clear whether or not the algorithm is really fast enough for this.

I understand the motivation of using SARCOS to test the capability of the algorithm to solve a motor task. However, since the paper's story has biology at the center, I think it would be much more convincing if the authors also took one of the tasks from one of their biological citations and replicated it with their algorithms. Even including the purely model-based task from Fiete et al, 2007 would make the argument that NP aids in songbird learning more than a passing remark.

Overall, I think a more clear story would be greatly beneficial to the paper and better determine its significance. If a central argument is that the brain *probably* does use regularized NP for motor learning, I would be skeptical and would be interested in further discussion of the biological literature; the citations Kornfeld et al, 2020 and possibly Bouvier et al, 2018 might support this line of argument if the results were better reflected upon in the paper. On the other hand, the story could be that the slowness of the algorithm, even in an idealized setting, demonstrates that the brain cannot be using it except as an auxiliary tool to a more powerful learning method.

---

> ### Author Response · Authors · 2022-08-02
> **Reply to Reviewer bpjP (1)**
>
> > One problem I have with the SARCOS figures is that there is no benchmark shown; everyone knows that an accuracy of 98\% on MNIST is acceptable, but since I'm not intimately familiar with the SARCOS task I would like to see either a SoTA MSE taken from the literature, or at least the MSE of the SGD-trained models alongside the NP-trained models for comparison. It is explained in the text that the target error was 5.0, but it is still not clear to me whether or not this is a good value in the grand scheme of algorithms which solve this task. Another way to make the error values easier to interpret would be to report \(R^2\) instead of MSE, but either way some kind of benchmark should be shown.
>
> Thank you for the helpful comments.
> When we trained a large network ($L_h=400$) with SGD until the training error effectively converges (10,000 epochs), we achieved MSE test error $\approx 1.6$, which is about three times better than the target we used.
> We are not aware of established state-of-the-art performance for the SARCOS task in the literature.
>
> In terms of the variance explained, we have $R^2 = 0.987$ at MSE = 5.0, and $R^2 = 0.996$ at MSE = 1.6.
> Thus, we believe MSE = 5.0 is an acceptable target value for a motor coordination task.
> We explained the variance-explained in the corresponding sentence in the main text.
>
> > I also think its worth noting that inverse problems (including SARCOS) are often solved innately by the brain. When a calf (or any quadruped) is born, they don't spend any number of epochs learning what torques are required to move their hooves from one place to the other, walking occurs immediately, suggesting that the capability was genetically encoded into the architecture, not learned. The authors could, of course, defend the claim that NP is used for learning more complex motor tasks, such as those performed by humans, but then it would also have to be clear whether or not the algorithm is really fast enough for this.
>
> Let us note that there are evidence suggesting developmental learning of motor coordination, although there is presumably a strong innate inductive bias.
>
> For instance, although calves manage to stand up within five minutes of birth as you mentioned, it takes up to a day for them to achieve adult-like locomotion. Quoting "The Birth and Survival of Wildebeest Calves (1977)",
> "Nevertheless, neonate wildebeest remained conspicuously unsteady for the first few h post partum. They constantly knocked against their mothers, stumbled over lying animals and other low obstacles, and every so often broke into a reeling gallop that looked certain to end in a nose dive, but seldom did...Such signs of unsteadiness diminished quickly, and were almost indiscernible to the human observer within a day of birth."
> Moreover, a recent work investigated postpartum development of piglet locomotion and found that it takes 2-8 hours for them to achieve symmetric locomotion (CV Hole et al., J Exp Biol, 2017), indicating the importance of learning for motor coordination.
>
> Regarding the latter remark on complex motor learning by humans, let us note that it often takes a huge number of trials for a human to learn a complex motor task. For example, it is estimated that $\sim 10^5$ trials are needed to achieve a professional golf stroke (M Meister, Curr. Opin. Neurobiol., 2022).
> We agree that it is important to study if this slow learning is consistent with perturbation-based learning, but we leave that for future work.

---

> > ### Comment · Reviewer_bpjP · 2022-08-03
> > **Clarification**
> >
> > > We explained the variance-explained in the corresponding sentence in the main text.
> >
> > Which line is that? I did not notice any clarifications in the text regarding SARCOS performance?

---

> > > ### Author Response · Authors · 2022-08-03
> > > **Re: clarification**
> > >
> > > It's in L269. The sentence now reads:
> > > "Under NP, the minimum training time required for reaching the target (here set to the test error, $\epsilon = 5.0$, *which corresponds to $R^2 = 0.986$*, roughly one hundredth of the initial error) depends only weakly on the hidden layer width."
> > > Sorry, it should be $R^2 = 0.986$, not $R^2 = 0.987$.

---

> > > > ### Comment · Reviewer_bpjP · 2022-08-08
> > > > **Re: Re: Clarification**
> > > >
> > > > Thank you for adding this; I think it does make the experimental results more clear. Although I think the results are overall solid and publishable, I also think they are highly specific and won't have high impact outside of this very particular area of bio-plausible learning, so I will retain my initial score of weak accept.

---

> > ### Comment · Reviewer_bpjP · 2022-08-03
> > **On the significance of the contribution**
> >
> > > Let us note that there are evidence suggesting developmental learning of motor coordination, although there is presumably a strong innate inductive bias.
> >
> > Your points on this are valid, so I improved my contribution score from 2 to 3.

---

> ### Author Response · Authors · 2022-08-02
> **Reply to Reviewer bpjP (2)**
>
> Thank you for your comments on the manuscript. Due to the word limit, we separated the reply into two sections.
>
> > My most central and concrete question is whether NP is slower or faster than the reinforcement algorithm. I got mixed messages from the figures and from the discussion section.
>
> NP is definitely slower than reinforcement learning. We made it clear both in the main results and the discussion (L279-281 and L317-319).
> Nonetheless, please note that the reinforcement learning algorithm we implemented relies on backpropagation, hence it is not biologically plausible.
>
> > In the abstract of Fiete et al, 2007, the authors state that their version of node perturbation "learns in a reasonable amount of time" and "Varying the number of neurons in HVC and RA has little effect on learning time." Do these 2007 results agree or disagree with the present results and why?
>
> We believe that their results are mostly consistent with our work.
> First, regarding the HVC size, their learning performance doesn't depend on the HVC due to their specific model setting.
> In their work, each HVC neuron, which is an input neuron, fires only once during a birdsong at a specific song segment. In addition, the feedback signal was provided to the network at each segment of a song.
> Because of this modular structure, they are effectively training a set of separate networks each corresponding to one segment of the song. Hence, the learning time doesn't depend on the total HVC size, though it expectedly depends on the HVC size per song segment.
>
> On the RA size dependence, in their model, the RA layer is the hidden layer, but because the connections from the RA layer to the output layer are fixed, the RA layer is effectively the output layer.
> Their numerical simulation demonstrated that the learning speed doesn't depend on the RA layer size. This is consistent with our finding of scalability against over-parameterization.
> However, their analysis is limited to a linear regression model; hence it wasn't clear if this observation is specific to their model setting in which the input activity is super-sparse and the target output is fixed.
> By contrast, here we analyzed the dynamics of one hidden layer neural network analytically, revealed when and why the network scales against over-parameterization in the hidden layer, and also uncovered its limitation in the presence of a model mismatch.
> We discussed the connection with this paper in the related work section.

---

### Official Review · Reviewer_up3q · 2022-07-05

**Rating:** 8
**Confidence:** 4
**Soundness:** 4 excellent
**Presentation:** 2 fair
**Contribution:** 4 excellent

**Summary:**

The authors present a neat analysis of the scalability and stability of the node perturbation algorithm, which is one of the popular bio-plausible credit assignment algorithms, for deep networks. Based on their analysis and inferences, they introduce a weight normalization trick that seems to alleviate the issues with the algorithm, albeit at the cost of adding a bias to the gradient estimates. In the first part of the paper, the authors use analytical tools to dissociate the effect of number of output nodes of the network from the number of perturbed nodes of the network to claim that node perturbation is scalable for deep linear networks. However, they demonstrate that the dynamics entails an instability in the weight norm. They validate these analytical results empirically in deep nonlinear networks and therefore establish a key result in training networks using node perturbation. They also show that the instablity is worse when there is noise in the labels (i.e. teacher noise). Finally, they demonstrate that the weight normalization trick can help stabilize the algorithm, albeit hurt performance by introducing a bias in the algorithm -- as evidenced by lower performance on the SARCOS and MNIST tasks.

**Questions:**

1. What is the dimensionality of $g_l$? I initially assumed that you used the numerator notation while writing vector calculus and therefore $g_l \in \mathcal{R}^{1 \times L_l}$, where $L_l$ is the dimensionality of $h_l$ or the number of neurons in $l^{th}$ layer. But from my understanding after reading the appendix, you assumed $g_l$ to be a column vector and therefore derived: $\frac{1}{\sigma}l(\tilde{x_K},x_0) - l(x_K,x_0) = \frac{1}{\sigma} \frac{\partial l}{\partial x_K} (\tilde{x_K} - x_K) = \sum_l \frac{\partial l}{\partial x_K} \frac{\partial x_K}{\partial h_l} \xi_l = \sum_l g_l^T \xi_l$. Is that correct? Perhaps add the dimensionality of the quantities in the appendix for clarity?
2. What does it mean to have lower linear dimensionality in the representations? Does the lower dimensionality underlie poor performance? This link was not clear from your description.
3. Re the weigh normalization scheme: Is it applied such that the weight normalization is the same at initialization? Can it be applied intermittently or as a regularizer? You mentioned why weight decay is not a solution to the instability problem but it wasn't clear if explicit weight normalization where the weights are allowed to grow from the initialization case could be a potential solution to the instability problem.

**Limitations:**

I think the authors do a commendable job in stating the limitations of their proposal but I feel it could be elaborated in the discussions. Specifically, how does the weight normalization introduce bias and its effect on the performance.

**Strengths And Weaknesses:**

Strength:
1. The paper is well motivated and tackles an important problem in the field. The analytical methods used in the paper are an important contribution towards understanding how node perturbation algorithms can be used to develop bio-plausible learning rules in deep networks.
2. The paper is theoretically strong and demonstrates via simulations how the analytical results hold in practice, as well as when nonlinearity is introduced in the network.
3. The weight normalization solution is a smart solution and a good integration of the inferences from the analytical framework. Furthermore, it also fits well with the homeostasis viewpoint wherein weights of a neuron are thought to conserve some quantity over time.

Weaknesses:
1. The presentation of the paper can be improved, particularly how the analytical results are presented. I felt a lot of the details of how the results were derived were buried in the appendix with limited reference to these considerations/assumptions in the main text. The authors could consider rewriting certain sections of the paper to better reflect the derivations of their theorems and in doing so, allow the reader to appreciate the contribution that this paper makes.
2. The weight normalization strategy could be better introduced, particularly how in adding a weight normalization step introduces bias in the weight update. It would be great if the authors could add elaborate on this bias and how a bigger hidden layer size contributes to a higher bias.
3. The metrics used in the paper make sense while analyzing the instablity of the node perturbation algorithm. However, it would be nice to report the accuracy/performance metrics on SARCOS and MNIST for SGD and weight-normalized NP. I feel this would be more complete and lay the foundation for further researchers (who could use weight-normalized NP as a baseline method).
4. The discussion about representation similarity is a bit sudden and lack compact presentation. Given that the merits of the paper lie in the stability analysis of the NP algorithm and proposing a possible workaround to that, I would suggest either cutting down on analyzing learned representations (although it is very important and an interesting direction) or fleshing it out more to convey the inferences better.

Overall, I feel that the paper presents a very thorough analysis of the NP algorithm and demonstrates a key feature in the dynamics. If the authors could improve the writing a bit, this work would appeal to a larger audience and could make a significant impact in the field.

---

> ### Author Response · Authors · 2022-08-02
> **Reply to Reviewer up3q**
>
> > 1. What is the dimensionality of $g_l$? I initially assumed that you used the numerator notation while writing vector calculus and therefore $g_l \in R^{1 \times L_l}$, where $L_l$ is the dimensionality of $h_l$ or the number of neurons in $l^{th}$ layer. But from my understanding after reading the appendix, you assumed $g_l$ to be a column vector and therefore derived: $\frac{1}{\sigma} l(\tilde{x}_K, x_0) - l (x_K, x_0) = \frac{1}{\sigma}\frac{\partial l}{\partial x_K} (\tilde{x}_K - x_K) = \sum_l \frac{\partial l}{\partial x_K} \frac{\partial x_K}{\partial h_l} \xi_l = \sum_l g_l^T \xi_l$. Is that correct? Perhaps add the dimensionality of the quantities in the appendix for clarity?
>
> Thank you for the helpful comments. Yes, that is correct. Throughout the paper, we defined all vectors as column vectors.
> We added explanations on the dimensionality in the Appendix (below Eq. 16 and Eq. 21).
>
> > 2. What does it mean to have lower linear dimensionality in the representations? Does the lower dimensionality underlie poor performance? This link was not clear from your description.
>
> Our analysis suggests that weight expansion underlies both lower linear dimensionality and poor performance.
> NP shows lower linear dimensionality than SGD mainly because the principal eigenvalue has a larger contribution under NP than SGD. This over-representation of the principal eigen-component is consistent with weight expansion, because the principal eigenvector of ReLU layer typically reflects the overall amplitude of the population activity, which goes up as the weights expand.
> The same weight expansion also induces instability in the learning dynamics, worsening the NP learning performance.
>
> We observed that, this low dimensionality can be observed even under a small learning rate in which NP performs comparative with SGD learning (Figs. S3C and D).
> It means that even when weight expansion is small enough to be harmless, you can still observe its signature in the linear dimensionality of the hidden layer activity.
> We made this point clear by adding Fig. S3C, which depicts the learning curves of NP and SGD under the same small learning rate.
> We also discussed the origin of low linear dimensionality in Appendix C.1.
>
> > 3. Re the weigh normalization scheme: Is it applied such that the weight normalization is the same at initialization? Can it be applied intermittently or as a regularizer? You mentioned why weight decay is not a solution to the instability problem but it wasn't clear if explicit weight normalization where the weights are allowed to grow from the initialization case could be a potential solution to the instability problem.
>
> We applied weight normalization at each update in a way that, the L2 norm of the incoming weights of each neuron remains the same with the initial L2-norm.
> Please note that the weight normalization can be rewritten as
> $$\boldsymbol{w}^k_i \rightarrow \boldsymbol{w}^k_i - \left( 1 - \frac{\lVert \boldsymbol{w}^k_i \rVert}{\lVert \boldsymbol{w}^k_i + \delta \boldsymbol{w}^k_i\rVert} \right) \boldsymbol{w}^k_i + \frac{\lVert \boldsymbol{w}^k_i \rVert}{\lVert \boldsymbol{w}^k_i + \delta \boldsymbol{w}^k_i\rVert} \delta \boldsymbol{w}^k_i$$
> meaning that the weight normalization can be interpreted as adaptive weight decay.
> The weight decay with a fixed decay ratio didn't work as effective, because that doesn't keeps the L2-norm constant even when the decay ratio is fine-tuned (black vs colored lines in Fig. S4F).
>
> Regarding the intermittent weight normalization, we observed that intermittent normalization slightly improves the convergence of NP compared to the normalization at every update. However, when the normalization is applied too infrequently, the error starts to show oscillatory behavior.
>
> > I think the authors do a commendable job in stating the limitations of their proposal but I feel it could be elaborated in the discussions. Specifically, how does the weight normalization introduce bias and its effect on the performance.
>
> The weight normalization introduces bias because it changes the direction of the weight update.
> In the presence of the weight normalization, the total weight change becomes $- \left( 1 - \frac{\lVert \boldsymbol{w}^k_i \rVert}{\lVert \boldsymbol{w}^k_i + \delta \boldsymbol{w}^k_i\rVert} \right) \boldsymbol{w}^k_i + \frac{\lVert \boldsymbol{w}^k_i \rVert}{\lVert \boldsymbol{w}^k_i + \delta \boldsymbol{w}^k_i\rVert} \delta \boldsymbol{w}^k_i$. Because of the first term, this update is no longer parallel to the direction of the gradient.
> We agree that it is important to study how this bias affects learning performance.
> However, analysis of a biased learning rule is generally much more difficult than analysis of a noisy learning rule, so we'd like to leave it for future work.

---

> > ### Comment · Reviewer_up3q · 2022-08-08
> > **Response to authors' response**
> >
> > I would like to thank the authors for the extemely high quality responses. Their responses have clarified a lot of my concerns and questions. I found the paper quite impressive and believe it will is a great contribution to the community. Personally, I would like to see the updated synaptic plasticity rule for weight normalization mentioned in the paper (or appendix). I believe it would help the community in extending the work to address some of the points laid out by the authors as future work.
> >
> > I should also add that some of the comments on reviewer 3nni resonated with my feelings about the "scalability" part of the work. But I was happy to see the theoretical framework presented by the authors in my initial review. Nevertheless, I will wait for the authors' response to the new comments from reviewer 3nni, specifically what is the main message that the authors would like a reader to understand from the new Cifar-10 experiments. I feel this is more a writing issue/concern and the authors could probably address this relatively easily, given their insights and understanding of the node perturbation algorithm. If not, please feel free to alert me of further complexities that I may have missed.
> >
> > Overall, I feel this is a strong submission and adds significant insight into the workings of node perturbation that is valuable to a portion of the NeurIPS community.

---

### Official Review · Reviewer_pkGH · 2022-07-05

**Rating:** 6
**Confidence:** 3
**Soundness:** 4 excellent
**Presentation:** 4 excellent
**Contribution:** 2 fair

**Summary:**

The paper considers the dynamics of training deep networks with node perturbation. The authors provide a detailed theoretical analysis of learning dynamics of node perturbation vs. SGD in linear networks with one hidden layer in the limit of infinite width. The analysis reveals that the input weight norm determines whether the loss increases or decreases during training with node perturbation, with large weight norm corresponding to unstable training. This motivates using weight normalization for node perturbation. Empircally, on linear and nonlinear networks, weight normalization stabilizes training with node perturbation.

**Questions:**

How does node perturbation with weight normalization perform in more complex datasets and architectures?

How might neuron-wise weight normalization be implemented biologically? Is there biological evidence for such a normalization?

Why is node perturbation a relevant model of biological learning to consider compared to other proposed biologically-plausible learning rules?

**Limitations:**

The authors address the limitations of their work in the discussion section. As they note, the utility of node perturbation is limited in the supervised setting, although its utility is more clear in a reinforcement learning setting. The authors may also want to comment on the potential applicability of their linear analysis to other settings; as they empirically find, the qualitative observations of the linear network extend to certain nonlinear networks. The authors may wish to specify in which settings these observations may not apply.

**Strengths And Weaknesses:**

 **Originality**
The work is original. The analysis on deep linear networks is novel and the insights from the experimental and theoretical results are not previously explored. To my knowledge, this paper is the first to provide a theoretical explanation for why weight normalization is useful for node perturbation.


**Quality**
The contributions are high quality. The theoretical analysis appears sound and the experiments are comprehensive. One potential drawback is that the experiments mainly consider simple datasets (SARCOS and MNIST) and mostly consider only one hidden layer networks (although a few experiments consider multilayer nonlinear networks). Investigating the performance of node perturbation in more challenging settings and more complex architectures could help justify the usefulness of the linear network analysis. However, this is not strictly necessary given that many of the main contributions of the paper are theoretical.


**Clarity**
The paper is well written. The mathematical notation is clear and the figures are well illustrated. However, many of the key and interesting theoretical results of the paper are in the supplementary material (particularly the mean-field dynamics of NP). It may help to provide a brief sketch of these results in the main paper.


**Significance**
The paper may have significance to researchers specifically studying node perturbation as a learning rule. However, its applicability to the field of biologically plausible learning appears more limited especially given that node perturbation does not appear to be empirically as effective as other biologically-plausible alternative learning rules. Moreover, the experimental results are limited to simple datasets and architectures. The significance of the paper could be significantly enhanced by exploring more complex settings and showing, for example, significantly improved performance of node perturbation when it is combined with weight normalization.

---

> ### Author Response · Authors · 2022-08-02
> **Reply to Reviewer pkGH**
>
> > How does node perturbation with weight normalization perform in more complex datasets and architectures?
>
> Thank you for the helpful comments. In newly added Figure S5, we investigated NP learning in a convolutional neural network solving CIFAR-10.
> As expected, vanilla NP learning becomes unstable in the middle of learning when the learning rate is too large (Fig. S5A).
> However, by adding weight regulation via weight decay, NP learning becomes stable even at a large learning rate (Fig. S5B), supporting the applicability of our results for complex networks and tasks.
> Here, we applied weight decay instead of weight normalization, because an implementation of the weight normalization in a convolutional network was somewhat tricky.
> We explained this result briefly at the end of section 5 in the main text.
>
>
> > How might neuron-wise weight normalization be implemented biologically? Is there biological evidence for such a normalization?
>
> Yes, we believe neuron-wise weight normalization is biologically well-grounded. Neuron-wise weight normalization can be rewritten as
> $$\boldsymbol{w}^k_i \rightarrow \boldsymbol{w}^k_i - \left( 1 - \frac{\lVert \boldsymbol{w}^k_i \rVert}{\lVert \boldsymbol{w}^k_i + \delta \boldsymbol{w}^k_i\rVert} \right) \boldsymbol{w}^k_i + \frac{\lVert \boldsymbol{w}^k_i \rVert}{\lVert \boldsymbol{w}^k_i + \delta \boldsymbol{w}^k_i\rVert} \delta \boldsymbol{w}^k_i$$
> Thus, we can interpret the weight normalization as adaptive weight decay.
> Previous experimental studies suggest that some form of adaptive weight decay is implemented in many types of neurons (see GG Turrigiano, Cell, 2008 for a review).
> The exact mechanism is not yet fully understood, but it is suggested that a neuron monitors its average firing rate, and down-scales its presynaptic weights when the firing rate becomes too high.
> We clarified this biological motivation for the weight normalization in Appendix C.1.
>
> > Why is node perturbation a relevant model of biological learning to consider compared to other proposed biologically-plausible learning rules?
>
> We think there are mainly three reasons why node perturbation remains relevant despite recent progress on the biologically-plausible learning rules.
> First, there are experimental evidence suggesting the existence of perturbation-driven learning, particularly in birdsong learning.
> In the songbird's brain, there is a region called LMAN which adds up variability to song production, but is also crucial for song acquisition (D Aranov et al., Science, 2008; F Ali et al., Neuron, 2013). Because of it, previous experimental and modeling works suggest that songbird learning is driven by node perturbation.
> By contrast, most of the biologically-plausible learning rules still lack experimental support.
>
> Secondly, synaptic plasticity in the brain is typically modulated by global error signals provided by neuromodulators. This is consistent with node perturbation, but not with many of biologically-plausible learning rules that rely on a tailored local error signals.
>
> Thirdly, neural activity in the brain is inherently stochastic. Node perturbation can naturally make use of this variability, while noise robustness of other biologically-plausible learning rules remains unclear.
>
> Another line of motivation for us to study node perturbation is to understand the bias-variance tradeoff in biologically plausible credit assignment mechanisms.
> Biologically plausible learning rules are inevitably either biased against SGD or noisier than SGD (see Fig. 2 of BA Richards et al., 2020, Nat Neurosci), but it remains unclear which learning rules the brain should use.
> We believe our study sheds light on when the brain shouldn't use a noisy update rule, and when it might be possible to make use of it.

---

> > ### Comment · Reviewer_pkGH · 2022-08-08
> > **Thank you for your response**
> >
> > I appreciate the authors for adding new experiments on CIFAR-10, additional improvements to the paper as well as fixing the experimental issues with the RL experiments. The authors' responses address some of my concerns: specifically, the biological grounding of node perturbation is more clear now.
> >
> > I have some remaining concerns regarding the CIFAR-10 experiments. First, it is important to note that the performance reported on CIFAR-10 in Figure S5 is relatively low even by the standards of biologically-plausible learning rules. Thus, we can't interpret node perturbation as being able to solve CIFAR-10 even with the appropriate choice of learning rate and weight decay. Moreover, given the finding that weight decay is not as effective as weight normalization, it would be more meaningful to evaluate weight normalization on CIFAR-10. Nevertheless, the finding that weight decay can stabilize learning even on CIFAR-10 is still valuable.
> >
> > Overall, I maintain my view that further that the paper's most valuable empirical contributions are on relatively simple datasets. However, the theoretical contributions are quite strong; I believe the paper's strengths outweigh its weaknesses.

---

### Official Review · Reviewer_3nni · 2022-07-06

**Rating:** 4
**Confidence:** 4
**Soundness:** 3 good
**Presentation:** 3 good
**Contribution:** 3 good

**Summary:**

The present paper analyzes the learning dynamics of Node Perturbation (NP), a biologically plausible learning algorithm used to train feedforward networks. Overall, the paper states a negative result about the unstability of NP due to weights diverging through learning, grounded in analytically tractable results obtained on linear models in the student-teacher setting carefully checked against numerical experiments. This analysis leads the authors to prescribe weight normalization to prevent this phenomenon. The predicted behavior in the linear regime is empirically observed in non-linear models on two training tasks, which validates the soundness of the aforementioned analysis.

More precisely:

- Section 3 introduces the NP algorithm, recalls that the resulting weight update provides an unbiased estimate of the SGD weight update and shows that the cosine similarity between these two weight updates for a given output layer scales as the inverse of the square-root of the size of this output layer (Eq. 8), suggesting that NP and SGD updates become nearly orthogonal for wide networks. Also, by comparing the covariance matrix of the weight updates for NP and SGD (Eq. 10), they want to highlight that NP is much noisier than SGD.

- Beginning of Section 4 recalls the minimum number of training steps needed to reach an error level for NP in the linear regression setting, and introduces a “deep linear model”, consisting of two linear layers, namely $y=W_2 \cdot W_1 \cdot x$. The rationale behind this choice is to analyze separately the impact of the number of perturbed units and that of the size of the output layer. This deep linear model is studied in the student-teacher setting, whereby the target is given by a teacher network consisting of a linear transformation with some additive Gaussian noise (if this noise is non-zero then the authors say there is a “mismatch” between the student and teacher networks).

- First paragraph of Section 4 shows the analytical NP learning dynamics in terms of the error $\epsilon$ (defined as the squared distance between student and teacher model) and the input weight norm $a=||W_1||$ through time (where time unit is a batch iteration), obtained in the large hidden layer, noiseless (i.e. no mismatch) limit. All the calculations leading to these results are provided in the Appendix. This analysis unveils two main results, summarized by Eq. 14. First, the weight norm grows monotonically increases through time. Second, there are two working regimes for NP, depending on the value of the learning rate used. If the learning rate is smaller than a critical threshold, the error converges to zero, if not the error decreases until the input weight norm a reaches a threshold (corresponding to the sign inversion of $\dot{\epsilon}$) wherefrom the error rises again. Hence the “instability”, which is caused by the input weight norm crossing a threshold. This theory is successfully checked against numerical simulations (Fig 1). Also, the minimal number of training steps required to reach an error level is analytically computed (Eq. 15) and numerically verified (Fig. 2), and highlights its weak dependency with the hidden layer size.

- Second paragraph extends the previous study to the case of mismatch (e.g. the labels given by the teacher network are noisy, $\sigma_t \neq 0$). It shows that the instability previously always happens, regardless of the learning rate value, the frontier at which it occurs is analytically derived (Eq. 16), and these results are checked numerically (Fig. 3A). There is still a “critical” learning rate threshold delimiting different scalings of the number of updates required to reach minimal error as a function of the learning rate (Fig 3B), a behavior which is in stark contrast with that of SGD (Fig 3C). Fig 4D shows that by normalizing the NP weight update, NP learning is stabilized, at the cost of a bias in the weight update which grows with the hidden layer size.

- In Section 5, the learning dynamics of NP are numerically analyzed on non-linear models on the MNIST and SARCOS training tasks. Again, the instability is observed and vanishes when normalizing the NP weight updates, suggesting that the previous analysis also holds in the non-linear setting. It is also shown that NP is up to 1000 times slower than SGD, which is partially explained by the fact that NP relies on a scalar rather than vector-valued error signal.

**Questions:**

- L.12: you haven’t yet defined “model mismatch” at this stage, I would remove it.
- L.24: “However, it remains unclear how robust these approximate learning rules are, or how they perform on large neural networks and complex tasks”. I would downweight this a little bit since you do not go beyond the MNIST task in this paper.
- L.90-106: I’m not sure the biasedness-noise tradeoff brings much added value here. It would free some space to add some more details about the teacher-student theory in the paper.
- L.493, Eq. 27b (Appendix): please detail the computation leading to $L_k + 2\delta_{kl}$. I could check by myself going on the internet to find the moments of a multivariate Gaussian, but I wouldn’t take for granted that any reader knows them by heart.
- L.506, Eq. 33 (Appendix): why does the tensor product between $\left(\sum_l \xi_k\xi_l^Tg_l – g_k\right)x_{k-1}^T$ and $(g_lx_{l-1}^\top - \langle g_l x_{l-1}^\top \rangle)$ disappear?
- L. 510, Eq. 36a (Appendix): I am really confused here. After you swap the second and third dimension, it seems to me, from Eq. 31, that $C_{kl}^{sgd}= \langle g_k g_l^\top \otimes x_{k-1}x_{l-1}^\top\rangle$, which is the second term of Eq. (36a). However, not only you distinguish $ C_{kl}^{sgd}$ from $\langle g_k g_l^\top \otimes x_{k-1}x_{l-1}^\top\rangle$ here, but also a factor 2 appears for no reason in front of $ C_{kl}^{sgd}$. Isn’t there a typo there? My take is that by adding up $ C_{kl}^{sgd}$ and $\langle g_k g_l^\top \otimes x_{k-1}x_{l-1}^\top\rangle$ you may obtain $ 2C_{kl}^{sgd}$.
- L.143: why do you call “deep” your linear model? It’s two layers deep. If you want to refer to the *width* of your linear model (since I acknowledge you study your model in the limit of a large number of hidden units), then use “width” instead of “depth”.
- L.149, Eq. 11: for completeness, it would have been useful to include the derivation of this result in the Appendix.
- **General important remark**: please do not abusively refer to equations or Figures in the Appendix, it hinders the fluid reading of the main.
- L.152: we do not understand here what you mean by “optimal learning rate”, i.e. how $\eta^*$ is precisely defined.
- L.549, Eq. 46 (Appendix): since the quantity $E\cdot x + \sigma_t z$ appears, it goes to suggest that $E$ is defined as $A – W_2\cdot W_1$, which is opposite in sign to Eq. 43.
- L.557 (Appendix): when you write that you “drop the higher-order terms”, I understand that you approximate the mean of a product of two quantities as the product of the means of these two quantities (at least this is how I can recover your result), is this correct? Please be more precise about it.
- L.561-567: very difficult to understand.
- While I can understand all the computations up until Eq. 57, from Eq. 58 onwards, the computations are no longer detailed. It’s very hard to follow because when you write $\approx$, you may perform two approximations at once: 1/ the mean of a product is the product of the means (eliminating “higher order terms”) 2/ the trace of a product is the product of the traces (Eq. 52).
- L.596, Eq. 71 (Appendix): please detail the computation.
- L.626, Eq. 82: with $0 < a_{init}<2/c_0$, you write $-a_{init}^3 < -(2/c_0)^3$, which is wrong ($x\to x^3$ increases on $\mathbb{R}^+$), so I am skeptical about the $\eta^*$ you obtain in Eq. 83.
- B.4 (appendix): these computations are not sufficiently detailed.
- L. 197: which saturation are you talking about? I can’t see the orange curve saturating on the Fig you refer to (Fig 2A).
- L.199-209: it’s very handwavy, and I do not understand the added value of this paragraph.
- Figure 5: this figure displays the accuracy while the others show the error. It would be easier to compare results across different figures if the same convention (all error curves, or all accuracy curves) is taken.

**Ethics Review Area:**

["I don’t know"]

**Limitations:**

My main concern is that this work,  titled "stability and **scalability** of NP learning", does not go beyond the MNIST task.

I think the most important contribution of this work is theoretical: there is an excellent match between theory and experiments as per the Figures shown in the main on linear models, which extend to non-linear models and gives a very simple insight as to how to use NP properly -- that is: by normalizing the NP weight updates. I think this simple result would have been even more compelling if for instance demonstrated on a 4-5 layers convolutional architecture trained by NP on CIFAR-10, the ideal situation being: without weight update normalization the model doesn't train, with weight update normalization it trains (even very slowly, but it does). Then, it could be concluded that the theoretical model proposed totally accounts for the unscalability of NP when normalization is not applied. As it stands, with MNIST being the most difficult task tested, it's harder to conclude the same.

In the bioplausible deep learning literature, the MNIST task alone as a benchmark to claim scalability is unsufficient to my eyes. However, it might be that the sole theoretical contribution of this work and MNIST benchmark abide by the standards of another community (e.g. statistical physics x ML) I may not be aware of, so my judgement might be biased by the fact I come from the bioplausible deep learning literature. Also, I think that the theoretical contributions should be better highlighted in the main.

I would recommend accept if:

1 - The authors ran a CIFAR-10 experiment with a 5 layers-deep convolutional architecture, observed the same kind of behavior as on MNIST and SARCOS.

2 - Computations in the appendix are more detailed than they are now -- see the questions section above.

**Strengths And Weaknesses:**

Strengths:
- The structure and the writing of the paper are clear. The figures are neat.
- The mean-field model derived in the linear student-teacher setting matches very well the experiments.
- The theoretical analysis seems to lead to a simple trick to unlock NP scalability.

Weaknesses:
- The derivations in the Appendix are not easy to follow, especially because multiple approximations are sometimes used without detailed steps (I will come back to this in the questions).
- The title raises big expectations that are not met: the term “scalability” calls for complex tasks beyond MNIST. This is a major limitation of the paper.

---

> ### Author Response · Authors · 2022-08-02
> **Reply to Reviewer 3nni (1)**
>
> > I would recommend accept if:
> > 1 - The authors ran a CIFAR-10 experiment with a 5 layers-deep convolutional architecture, observed the same kind of behavior as on MNIST and SARCOS.
> > 2 - Computations in the appendix are more detailed than they are now -- see the questions section above.
>
> Thank you for the detailed comments on the manuscript.
> On the first point, we applied node perturbation (NP) to deep convolutional networks solving CIFAR-10 as you requested. The table below compares the performance of the vanilla NP and SGD.
>
> | Network   | NP (50 epochs) | SGD (50 epochs) | NP (5000 epochs) | SGD (5000 epochs) |
> |---|---|---|---|---|
> | Fully-connected | 38.7 | 45.9 | 50.4 | 52.3 |
> | Small conv-net | 33.9 | 61.6 | 48.2 | 62.6 |
> | Large conv-net | 13.6 | 70.6 | 30.9 | 72.5 |
>
> Here, the small conv-net is a three-hidden layer convolutional network with 64 channels each, and the large conv-net is the seven-layer all convolutional network from Springenberg et al., 2014. We optimized the learning rate for each architecture and training time.
> As shown above, even without any weight normalization, NP learns the task to some extent, but learning is slow. Especially in convolutional networks, 5000 epochs of training was not enough to achieve the performance SGD achieved with 50 epochs.
> Because of this slow learning process, it is difficult to perform a detailed analysis of NP learning behavior in convolutional network solving CIFAR-10; thus we focused on the issue of stability.
>
> As shown in Figure S5A in the Appendix, under a large learning rate, NP dynamics indeed becomes unstable in the middle of learning (green and lime lines).
> However, by regulating the weight norm with weight decay, we managed to stabilize NP learning dynamics (Fig. S5B; light green vs black lines).
> Thus, our theoretical prediction on NP stability holds qualitatively even in a convolutional neural network solving CIFAR-10.
>
> Please also note that, in terms of scalability, we only claim scalability against over-parameterization, but not against task complexity. We clarified it in the second paragraph of the discussion: "Though we found the minimum training time of NP is relatively robust against over-parameterization, its scalability against complex supervised learning tasks is clearly limited due to this slowness."
>
> Regarding your second point, we expanded Appendix based on your comments. Please see our replies to your questions below for the details.
>
> ---
> > L.12: you haven’t yet defined “model mismatch” at this stage, I would remove it.
>
> We specified what we mean by the word "model mismatch".
> The sentence now reads "However, unlike stochastic gradient descent, when there is a model mismatch *between the student and teacher networks*, node perturbation is always unstable."
>
> > L.24: “However, it remains unclear how robust these approximate learning rules are, or how they perform on large neural networks and complex tasks”. I would downweight this a little bit since you do not go beyond the MNIST task in this paper.
>
> We removed "complex tasks" from the sentence.
>
> > L.90-106: I’m not sure the biasedness-noise tradeoff brings much added value here. It would free some space to add some more details about the teacher-student theory in the paper.
>
> We shortened the paragraph, but kept it in the manuscript because we believe this paragraph put our work in a larger context of the bias-variance tradeoff in learning algorithms.

---

> > ### Comment · Reviewer_3nni · 2022-08-08
> > **Feedback on CIFAR-10 experiments**
> >
> > Hi, thanks for performing these extra experiments!
> > However I still have questions there:
> >
> > - For me the expected logics are as follows: 1/ vanilla NP works for small learning rates, but is subsequently slow 2/ if learning rate is increased, NP becomes instable (as predicted by the theory) 3/ to accommodate larger learning rates and guarantee learning stability **as well as learning within a feasible time** (which is the initial goal of increasing the learning rate, right?), weight normalization or weight decay is added. However, what you are doing here is a little bit dubious. When looking at the new figure in Appendix (Fig. S5), I do observe 1/ and 2/ -- for learning rates $> 1e-3$, learning becomes unstable. However, to check 3/, to generate the right panel of Fig. S5, you say you set the learning rate at $2e-4$, while I would have expected you set it at $2e-3$. I guess this is a typo and you wanted to write $2e-3$ instead (since the the black line corresponds to the green line in the left panel).
> >
> > - I do not understand your choice for the extra simulations you performed on CIFAR-10: why did you run experiments on _vanilla_ NP instead of NP with weight decay of weight normalization? Your analysis in Fig 5S shows that you can use a larger learning rate if you use weight decay, why didn't you leveraged this, especially because you say in your answer that vanilla NP is too slow? Unless I have misunderstood something here, this does not make sense to me.
> >
> > - I'm not sure how to conclude clearly about the CIFAR-10 experiments and overall the take-away of the paper. My initial thinking was along these lines: "NP does not scale because of a fundamental double-edged issue: either it is too slow, but if learning rate is increased learning becomes unstable because of weight magnitude divergence. If you want to scale NP to deeper architectures, prevent this divergence by weight normalization to allow for larger learning rates". However, while you show on Fig 5S that you can indeed "stabilize" learning for lr=$2e-3$ (I suppose this is this value that you used and not $2e-4$) by employing weight decay, you do not show the results you obtain on CIFAR-10 by doing so! So it's difficult for me to conclude that your theoretical analysis extend to CIFAR-10 with the elements I have here. Alternatively, could it be that the conclusion of the study is "NP is doomed to be unscalable because either learning is too slow, or it is unstable for the fundamental reason we identified, and that weight normalization is not enough to use NP in the large learning rate regime and prevent weight magnitude divergence"? For me the conclusion is really not clear at this stage.

---

> > > ### Author Response · Authors · 2022-08-09
> > > **Reply to Feedback on CIFAR-10 experiments**
> > >
> > > Thank you for your comments.
> > >
> > > 1. The legend of Fig. S5 indeed contains a typo. It should be $2\times10^{-3}$, not $2 \times 10^{-4}$. We fixed the typo in the figure legend.
> > >
> > > 2. We have now implemented weight normalization in convolutional networks and conducted a few long training, as well as in networks with weight decay.
> > > Our preliminary results indicate that, even with weight normalization, NP won't reach SGD-level performance in CIFAR-10 ($\sim 60 \\%$ in our network) after a few thousand epochs of training.
> > > This is consistent with our finding in one hidden layer linear networks: the larger the network is, the more bias is caused by the weight normalization (Fig. 3D).
> > >
> > > 3. Regarding supervised image classification tasks, our conclusion is the latter: Node perturbation is not a feasible option for these tasks.
> > > This is mainly supported by our finding of its instability and observed slowness. Although weight normalization stabilizes the learning process, it also induces bias in the weight update; hence impairing convergence. Please note that, even when NP learns a task successfully with weight normalization, it still takes a hundred times more iterations than SGD (Fig. 5BC).
> > > This conclusion is stated both in the introduction (L67: "NP is too slow to be practical in supervised image recognition tasks.") and the discussion (L315: "...its scalability against complex supervised learning tasks is clearly limited due to this slowness.").
> > > We have now also clarified our conclusion with regards to CIFAR-10 in both the introduction and the main result.
> > > L60-61 in the introduction now reads "We also confirmed the stabilization of learning dynamics by weight regularization in convolutional neural networks solving the CIFAR-10 task, *but it also impairs performance.*"
> > > And the last line of section 5 (L302-L304) goes "Lastly, we applied NP to a convolutional network learning CIFAR-10, and found that weight regularization stabilizes NP learning dynamics (Fig. S5). *However, it only achieves a low accuracy (< 50\%) after a few thousand epochs of training due to its bias and slowness.*" \
> > > &emsp; Nonetheless, we believe it is an overstatement to claim that NP is a lost cause as a candidate credit assignment mechanism in the brain.
> > > This is based on our two positive findings on NP: The minimum training time of NP is relatively robust against over-parameterization, and a basic technique like weight normalization significantly improves the learning performance of NP at least in simple tasks.
> > > Although these properties are not enough to scale NP to CIFAR-10, it may still be useful in other types of credit assignment problems in the brain.
> > > For instance, in motor learning, the weight updates often need to rely on noisy reinforcement signals, which makes learning slow regardless of the learning algorithm.
> > > It is also known that motor learning is typically a time-consuming process. It takes several months for a songbird to learn a song, and a decade for a human to master a professional golf swing.
> > > For these reasons, we simply stated that we revealed the limitation and potential of NP, both at the end of the abstract (L17-19) and the beginning of the discussion (L306-307).

---

> ### Author Response · Authors · 2022-08-02
> **Reply to Reviewer 3nni (2)**
>
> > L.493, Eq. 27b (Appendix): please detail the computation leading to $L_k + 2 \delta_{kl}$. I could check by myself going on the internet to find the moments of a multivariate Gaussian, but I wouldn’t take for granted that any reader knows them by heart.
>
> Taking the expectation over $\boldsymbol{\xi}$, we have
> $$\begin{align}
> \langle \boldsymbol{\xi}\_l \boldsymbol{\xi}\_k^T \boldsymbol{\xi}\_k \boldsymbol{\xi}\_m^T \rangle
> &= \delta\_{lm} \langle \boldsymbol{\xi}\_l \boldsymbol{\xi}\_k^T \boldsymbol{\xi}\_k \boldsymbol{\xi}\_l^T \rangle
> \nonumber \\\\
> &= \delta\_{lm} ([1 - \delta\_{kl}] \langle \boldsymbol{\xi}\_k^T \boldsymbol{\xi}\_k \rangle \langle \boldsymbol{\xi}\_l \boldsymbol{\xi}\_l^T \rangle + \delta\_{kl} \langle \boldsymbol{\xi}\_k \boldsymbol{\xi}\_k^T \boldsymbol{\xi}\_k \boldsymbol{\xi}\_k^T \rangle)
> \nonumber \\\\
> &= \delta\_{lm} ([1-\delta\_{kl}] L\_k \boldsymbol{I}\_k + \delta\_{kl} [L\_k + 2] \boldsymbol{I}\_k )
> \nonumber \\\\
> &= \delta\_{lm} (L\_k + 2 \delta\_{kl}) \boldsymbol{I}\_k,
> \end{align}
> where $\boldsymbol{I}\_k$ is the size $L\_k$ identity matrix.
> Please note that, $(\mu, \nu)$-th element of matrix
> $\langle \boldsymbol{\xi}\_k \boldsymbol{\xi}\_k^T \boldsymbol{\xi}\_k \boldsymbol{\xi}\_k^T \rangle$ is given as
>
> $$\begin{align}
> [\langle \boldsymbol{\xi}\_k \boldsymbol{\xi}\_k^T \boldsymbol{\xi}\_k \boldsymbol{\xi}\_k^T \rangle ]\_{\mu \nu}
> &= \delta\_{\mu \nu} \sum\_{\rho=1}^{L\_k} \langle \xi^k\_{\mu} \xi^k\_{\rho} \xi^k\_{\rho} \xi^k\_{\mu}\rangle
> \nonumber \\\\
> &= \delta\_{\mu\nu} \sum\_{\rho=1}^{L\_k} ( [1 - \delta\_{\mu \rho}] \langle (\xi^k\_{\mu})^2 \rangle \langle (\xi^k\_{\rho})^2 \rangle + \delta\_{\mu \rho} \langle (\xi^k\_{\mu})^4 \rangle )
> \nonumber \\\\
> &= \delta\_{\mu \nu} (L\_k + 2).
> \end{align}$$
> We added this detailed derivation below the corresponding equation in the Appendix (Eqs. 26 and 27).
>
>
> > L.506, Eq. 33 (Appendix): why does the tensor product between $(\sum_l \xi_k \xi_l^T g_l - g_k) x_{k-1}^T\) and \((g_l x_{l-1}^T - \langle g_l x_{l-1}^T \rangle)$ disappear?
>
> It disappears, because by taking expectation over $\xi$, we have
> $$\left\langle (\sum_l \xi_k \xi_l^T g_l - g_k) x_{k-1}^T \otimes (g_k x_{k-1}^T - \langle g_k x_{k-1}^T \rangle) \right\rangle
> = \langle (g_k - g_k) x_{k-1}^T \otimes (g_k x_{k-1}^T - \langle g_k x_{k-1}^T \rangle) \rangle = 0.$$
>
>
> > L. 510, Eq. 36a (Appendix): I am really confused here. After you swap the second and third dimension, it seems to me, from Eq. 31, that $C^{sgd}\_{kl} = \langle g_k g_l^T \otimes x_{k-1} x_{l-1}^T \rangle$, which is the second term of Eq. (36a). However, not only you distinguish $C^{sgd}\_{kl}$ from $\langle g\_k g\_l^T \otimes x\_{k-1} x\_{l-1}^T \rangle$ here, but also a factor 2 appears for no reason in front of $C^{sgd}\_{kl}$. Isn’t there a typo there? My take is that by adding up $C^{sgd}\_{kl}$ and $\langle g\_k g\_l^T \otimes x\_{k-1} x\_{l-1}^T \rangle$ you may obtain $2 C^{sgd}\_{kl}$.
>
> Thank you for pointing it out. The second line indeed contains a typo. The second line should be
> $C^{sgd}\_{kl} + [\langle g\_k x\_{k-1}^T\rangle \otimes \langle g\_l x\_{l-1}^T \rangle ]\_{2 \leftrightarrow 3} + \delta\_{kl} \langle \sum\_m \lVert g\_m \rVert^2 I\_k \otimes x\_{k-1} x\_{k-1}^T \rangle$, where
> $[\cdot]\_{2 \leftrightarrow 3}$ indicates switching of the second and the third dimensions.
> Applying $\langle g\_k x\_{k-1}^T\rangle \approx 0$ to this equation, we get the third line of the equation. We fixed the typo in the manuscript.
>
>
> > L.143: why do you call “deep” your linear model? It’s two layers deep. If you want to refer to the width of your linear model (since I acknowledge you study your model in the limit of a large number of hidden units), then use “width” instead of “depth”.
>
> We changed "deep linear networks" in the abstract to "deep linear networks *with one hidden layer*".
> Please note that int the main text and figure legends, we denoted the linear model we analyzed as "one hidden layer linear networks".
>
>
> > L.149, Eq. 11: for completeness, it would have been useful to include the derivation of this result in the Appendix.
>
> We added a short derivation of the minimum training time of NP and SGD in linear regression in Appendix B.6.
>
>
> > General important remark: please do not abusively refer to equations or Figures in the Appendix, it hinders the fluid reading of the main.
>
> Thanks for the remark. Due to the tight page constraint, we had to move most of the equations and some of the figures into the Appendix; hence we needed to refer to figures and equations in the Appendix frequently. We believe the main text is written in a way that you can understand it without referring to the Appendix. If there are specific points, please let us know.

---

> ### Author Response · Authors · 2022-08-02
> **Reply to Reviewer 3nni (3)**
>
> > L.152: we do not understand here what you mean by “optimal learning rate”, i.e. how $\eta^*$ is precisely defined.
>
> We defined the optimal learning rate as the learning rate that minimizes the learning time. In Figure S1, the optimal learning rate is the point at which the training time curve touches the horizontal dotted line.
> We clarified the definition of the optimal learning rate in the corresponding sentence. The sentence now goes: However, the optimal learning rate -- *the learning rate that minimizes the training time* -- is smaller under NP than SGD, so the minimum training time becomes longer (Fig. S1; *here, the optimal learning rates are the points at which the curves touch the horizontal dotted lines*).
>
> > L.549, Eq. 46 (Appendix): since the quantity $E \cdot x + \sigma z$ appears, it goes to suggest that $E$ is defined as $E = A - W_2 W_1$, which is opposite in sign to Eq. 43.
>
> Here, we flipped the sign of $\sigma z$, not $E$. Because $z$ is a zero-mean Gaussian random variable, $y^* = Ax + \sigma z$ can be replaced with $y^* = Ax - \sigma z$, as long as the subsequent equations are consistent. We clarified it below Eq. 46.
>
> > L.557 (Appendix): when you write that you “drop the higher-order terms”, I understand that you approximate the mean of a product of two quantities as the product of the means of these two quantities (at least this is how I can recover your result), is this correct? Please be more precise about it.
>
> "Dropping the higher order term" is an approximation method in which one ignores higher-order correlation of Gaussian random variables.
> In our problem, by taking expectation over Gaussian random variables using Eq. 27, we have
> $$\begin{align}
> &\langle tr[ (W_2 \xi_1 \xi_1^T \xi_1 \xi_1^T W_2^T + \xi_2 \xi_1^T \xi_1 \xi_2^T) (E x x^T x x^T E^T + \sigma_t^2 z x^T x z^T)] \rangle
> \nonumber \\\\
> &= tr[([L_h+2] W_2 W_2^T + L_h I_y) ([L_x+2]EE^T + \sigma_t^2 L_x I_y)].
> \end{align}$$
> Using $L_h,L_x \gg 1$, we can approximate $L_h+2 \approx L_h$ and $L_x+2 \approx L_x$, then we get the third line of Eq. 51. Because "+2" factor comes from higher-order moments, this type of approximation is commonly referred to as "dropping the higher-order terms" in the physics community.
> We explained the details of this approximation below the equation.
>
> > L.561-567: very difficult to understand.
>
> We added a reference to Appendix B.5 in L 583 (prev. L563), where the details of approximation and its limitation are discussed.
>
> > While I can understand all the computations up until Eq. 57, from Eq. 58 onwards, the computations are no longer detailed. It’s very hard to follow because when you write $\approx$, you may perform two approximations at once: 1/ the mean of a product is the product of the means (eliminating “higher order terms”) 2/ the trace of a product is the product of the traces (Eq. 52).
>
> While it is true that we used both approximations (dropping the higher-order terms and replacing the trace of a product with the product of the traces) in Eqs. 59 and 60, in all equations, we first dropped the higher-order terms, then approximated the trace of a product with the product of the traces.
> In Eq. 60, the right-hand side of the first lines are the result of dropping the higher-order terms, and the second lines are the result of applying Eq. 52 or Eq. 57. We explained it below Eq. 59.
>
> > L.596, Eq. 71 (Appendix): please detail the computation.
>
> We have $\alpha_{init} = \frac{1}{L_x} \sum_{i=1}^{L_h} \sum_{j=1}^{L_x} (W^1_{ij})^2 \approx L_h \langle w^2 \rangle_{w \sim N(0,2/(L_x+L_h))} = \frac{2 L_h}{L_x + L_h}$.
> In the second equality, we replaced the average over $i$ and $j$ with the expectation over Gaussian variable $w \sim N(0,2/(L_x+L_h))$ using $L_x, L_h \gg 1$. The rest of the terms follow from similar calculations.
> To be precise, we replaced $=$ in the equations with $\approx$, and added an explanation below the equations.
>
> > L.626, Eq. 82: with $0 < a_{init} < 2/c_o$, you write $-a_{init}^3 < -(2/c_o)^3$, which is wrong ($x \to x^3$ decreases on $R^+$), so I am skeptical about the $\eta^*$ you obtain in Eq. 83.
>
> This inequality is correct, because it works on $f(a) = -a^3/3 + a^2/c_o$ not on $-a^3/3$.
> Please note that $f(a) = -a^3/3 + a^2/c_o$ is a monotonically increasing function within $0 < a < 2/c_o$, thus $0 < f(a) < f(2/c_o)$.
>
> > B.4 (appendix): these computations are not sufficiently detailed.
>
> We expanded the derivation of the minimum training time. We kept the first half of the subsection concise because the calculations are effectively a repetition of the same calculations for NP.

---

> > ### Comment · Reviewer_3nni · 2022-08-08
> > **Feedback (part II)**
> >
> > OK!

---

> ### Author Response · Authors · 2022-08-02
> **Reply to Reviewer 3nni (4)**
>
> Due to the word limit, we separated the reply into four sections. Please read from (1) to (4).
>
> Thanks again for the detailed comments. These comments helped us to improve the clarity of the manuscript.
>
>
> > L. 197: which saturation are you talking about? I can’t see the orange curve saturating on the Fig you refer to (Fig 2A).
>
> The saturation happens only at the very large hidden layer size $L_h$, which is captured by the orange line in Fig. S2A in the Appendix.
> Please note that, even under a small $L_h$, $L_h$ dependence of the training time is very weak (Fig. 2A).
>
> > L.199-209: it’s very handwavy, and I do not understand the added value of this paragraph.
>
> We believe this intuitive explanation helps more neuroscience-oriented readers, though admittedly this is rather hand-wavy.
>
> > Figure 5: this figure displays the accuracy while the others show the error. It would be easier to compare results across different figures if the same convention (all error curves, or all accuracy curves) is taken.
>
> We added a panel depicting the learning curves measured in MSE error in log-log scale as Fig. S4G in Appendix.
> In Figure 5, we plotted the performance using the one-hot accuracy as is commonly done for the MNIST dataset.

---

> > ### Comment · Reviewer_3nni · 2022-08-08
> > **Feedback (part I)**
> >
> > OK!

---

### Author Response · Authors · 2022-08-02
**Reply to the reviewers**

Thank you all for the helpful comments. Please find the attached revised manuscript and point-to-point replies to the comments below.

Major changes to the manuscripts are

- We added new numerical results on the effect of weight regulation on NP in a deep convolutional network solving CIFAR-10 (Fig. S5).
We found that weight regulation by weight decay indeed stabilizes the NP learning dynamics, although NP still performs poorly compared to SGD.

- We added a brief summary of the derivation of the mean-field learning dynamics in section 4, to provide a better intuition on the analysis. We also expanded the appendix, by adding more details on weight normalization, linear dimensionality estimation, and the linear network analysis.

- We realized that the original numerical experiment of the reinforcement algorithm contains an error, which we now fixed (solid green lines in Fig. 4E and Fig. 5B).
It revealed that the performance of the reinforcement algorithm is closer to SGD than our original results suggest. It does not affect our main results on the stability and scalability of NP.

---

### Meta-Review · Area_Chair_vsDc · 2022-08-25

**Recommendation:** Accept
**Confidence:** Certain

**Metareview:**

Authors theoretically analyze and numerically verified statistical properties of node perturbation which is one of the more biologically plausible but slower learning rule. Authors show both the benefits and limitations of the naive node perturbation in terms of learning trajectory. In particular, node perturbation is unstable under practical regimes. They propose a biologically plausible weight normalization scheme which overcomes some of the limitations of the naive version (but introduces some bias). This work advances the theoretical understanding with significant contribution to the neuroscience of learning. The expert reviewers agree that the work is original, clear, and of high quality. I suggest revising the title to indicate both the negative and positive sides of the analysis, perhaps "On the stability and scalability of Node Perturbation Learning" would be better.

**Award:**

No

---

### Decision · Program_Chairs · 2022-09-14

Accept